# Enhancing sorption kinetics by oriented and single crystalline array-structured ZSM-5 film on monoliths

Junfei Weng [1], Chunxiang Zhu [1], Binchao Zhao [1], Wenxiang Tang[1], Xingxu Lu[1], Fangyuan Liu [1], Mudi Wu [1], Yong Ding[2] & Pu-Xian Gao [1] ✉

To enhance the reaction kinetics without sacrificing activity in porous materials, one potential solution is to utilize the anisotropic distribution of pores and channels besides enriching active centers at the reactive surfaces. Herein, by designing a unique distribution of oriented pores and single crystalline array structures in the presence of abundant acid sites as demonstrated in the ZSM-5 nanorod arrays grown on monoliths, both enhanced dynamics and improved capacity are exhibited simultaneously in propene capture at low temperature within a short duration. Meanwhile, the ZSM-5 array also helps mitigate the long-chain HCs and coking formation due to the enhanced diffusion of reactants in and reaction products out of the array structures. Further integrating the ZSM-5 array with $Co_3O_4$ nanoarray enables comprehensive propene removal throughout a wider temperature range. The array structured film design could offer energy-efficient solutions to overcome both sorption and reaction kinetic restrictions in various solid porous materials for various energy and chemical transformation applications.

Solid porous materials[1], such as activated carbon, metal oxides[2], zeolites[3], and metal-organic frameworks (MOF)[4], have been extensively applied in catalysis[5–7], membrane separation[8], adsorption/storage[9], and sensing[10]. Their structural and functional advantages, such as high surface area, ordered structure, adjustable chemical composition, and diffusion selectivities over guest components, have enabled the porous materials to exhibit remarkable versatilities in the adsorption, catalytical activity, separation, and storage for various gases and liquids.

In the past decades, extensive efforts have been made to break through the porous materials performance, such as the maximum catalytic conversion efficiency, adsorption uptake, and separation factors. Comparably, less attention has been focused on addressing the kinetic restrictions, which could play more significant roles in transient reactions or short process durations that demand high energy-efficiency and fast diffusion kinetics. For example, low-temperature hydrocarbons (HCs) traps such as zeolites are needed during the automotive cold-start period before the three-way catalysts (TWC) or diesel oxidation catalysts (DOC) are thermally activated to effectively remove unburnt HCs[11]. The transition stage of a cold-start period is usually as short as less than 3 minutes, during which unburnt HCs could be released regardless of the HCs uptake performance in the zeolite trap[12,13]. This underscores the importance of searching and finding appropriate means to address such kinetic constraints.

It is noted that a potential solution to such a kinetic-limitation challenge may be present via manipulating the orientation and hierarchical structuring of various porosity that may exist in nanostructure assemblies or porous materials. For instance, in the past decade, a series of metal-oxide nanoarray-based monolithic catalysts and reactors have been demonstrated for various heterogeneous reactions ranging from oxidation of CO, $CH_4$, HCs, NO, and soot, to selective reduction of $NO_x$ and $CO_2$ hydrogenation[14–19]. Their catalytic performance was demonstrated to be controllable by tuning the porosity and orientation of nanoarray systems. Lu et al.[20] also quantified and

[1]Department of Materials Science and Engineering & Institute of Materials Science, University of Connecticut, Storrs, CT 06269, USA. [2]School of Materials Science and Engineering, Georgia Institute of Technology, Atlanta, GA 30332, USA. ✉e-mail: puxian.gao@uconn.edu

validated the superior mass transport advantage in these nanoarray structured monoliths, with much lower internal diffusion limitations than the traditional powder-washcoated counterpart.

Similarly, the porous materials such as zeolites can display anisotropic distribution of pores and channels that could also be designed and tailored to fit specific application requirements[21–25]. For example, ZSM-5 (MFI framework) is a classic and widely-used type of zeolites with sinusoidal channels along *a*-axis and straight channels along *b*-axis. As illustrated in Fig. 1a, continuous *b*-axis oriented MFI thin films have been desired for separation of chemical molecules (e.g., xylene isomers) due to a faster diffusion rate through the short and straight channels along *b*-axis[26–29]. Meanwhile, MFI films also selectively contribute to the catalytic activity, lifetime, and product selectivity through their zigzag *a*-orientation channels[30,31]. Although the guest species (e.g., propene) was proved to be captured on both sinusoidal and straight channels and their intersections in ZSM-5 crystals[13,32,33], the orientation control strategy is seldomly utilized for adsorption application especially in the dynamics aspect.

In this work, we propose to vertically grow a *c*-oriented ZSM-5 film on monolithic support with array structure (Fig. 1b) as a practical and convenient approach to improve the reaction kinetics towards HCs adsorption at low temperature. The hypothesis is that, together with the macroporosity engineered in the array structure[34,35], the preferential growth of *c*-oriented ZSM-5 nanocrystals would maximize the exposure of pores and channels along *a*- and *b*-orientations laterally, and thus the adsorption process could be facilitated. Meanwhile, a high concentration of acid sites engineered in the array-structured ZSM-5 film could contribute to a high HCs adsorption uptake.

## Results and discussion
### Catalyst preparation and characterization
We prepared the *c*-axis oriented ZSM-5 film on cordierite honeycomb substrates using the secondary growth method, as detailed in the Methods. Two types of ZSM-5 films at Si/Al ratios of 80 and 20,

denoted as "Z-80-F" and "Z-20-A," were synthesized on the channeled cordierite monolith with the assistance of pre-deposited silicalite-1 seeds[36,37]. The ZSM-5 crystals are identified on both samples in the X-ray diffraction (XRD) pattern as shown in Supplementary Fig. 1. Upon the seeded monolith (Supplementary Fig. 2) is exposed to the synthesis solution at a Si/Al ratio of 80 at 180 °C, a dense film is formed on the channel surface (Z-80-F, Fig. 2a) with highly intergrown crystal distribution, similar to the reported conventional film morphology[38]. However, when the Si/Al ratio decreases from 80 to 20, the ZSM-5 film morphology changes dramatically (Supplementary Fig. 3), with the loading ratio decreasing from 26.0 wt.% to 19.2 wt.% (Supplementary Table 1). Instead of the crystal intergrowth, Z-20-A presents a unique array structure composed of vertically oriented ZSM-5 nanorods with well-preserved single crystal individually (Fig. 2b). Each ZSM-5 nanorod crystal is ~200-400 nm in diameter and ~3 µm in length. This array structure is further confirmed by the bright-field transmission electron microscopy (TEM) image in Fig. 2d. The high-resolution TEM images and inset selected area electron diffraction (SAED) patterns in Fig. 2f indicate that the single-crystalline ZSM-5 nanorod in Z-20-A grows along [002] (*c*-axis orientation).

Samples synthesized at 180 °C for different durations are prepared to investigate the growth habits of conventional continuous ZSM-5 film and array-structured ZSM-5 film, as summarized in the time-in-series SEM image galleries in Supplementary Fig. 4 and weight change in Supplementary Fig. 5. Unlike the crystal intergrowth that quickly develops in the formation of a continuous ZSM-5 film in Z-80-F, the crystal individuality is preserved at the early growth stage of Z-20-A. The difference in film morphology and loading ratio in Z-80-F and Z-20-A might be strongly influenced by the Al ion concentration evolution in the synthesis solution. During the film growth, the negative charge resulting from the replacement of Si by Al in the tetrahedral units of ZSM-5 crystal could be accumulated on the crystal surface[39]. Since the MFI precursors in the synthesis solution are also negatively charged, the transport and attachment of nutrition particles to the

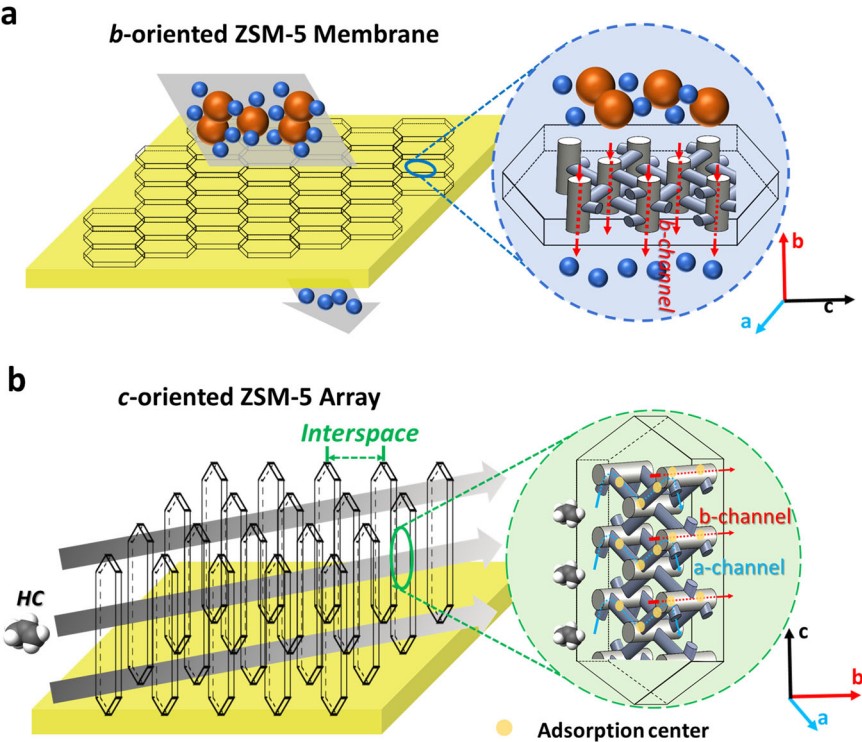

**Fig. 1 | The schematic illustration of ZSM-5 films with different crystal orientations. a** *b*-axis oriented ZSM-5 film desired for separation and corresponding *b*-axis molecular channel distribution within. **b** The proposed *c*-axis oriented ZSM-5 single crystal array film with exposure of molecular channels along both *a*- and *b*-axes to provide adequate active centers for adsorption of small molecules such as propene.

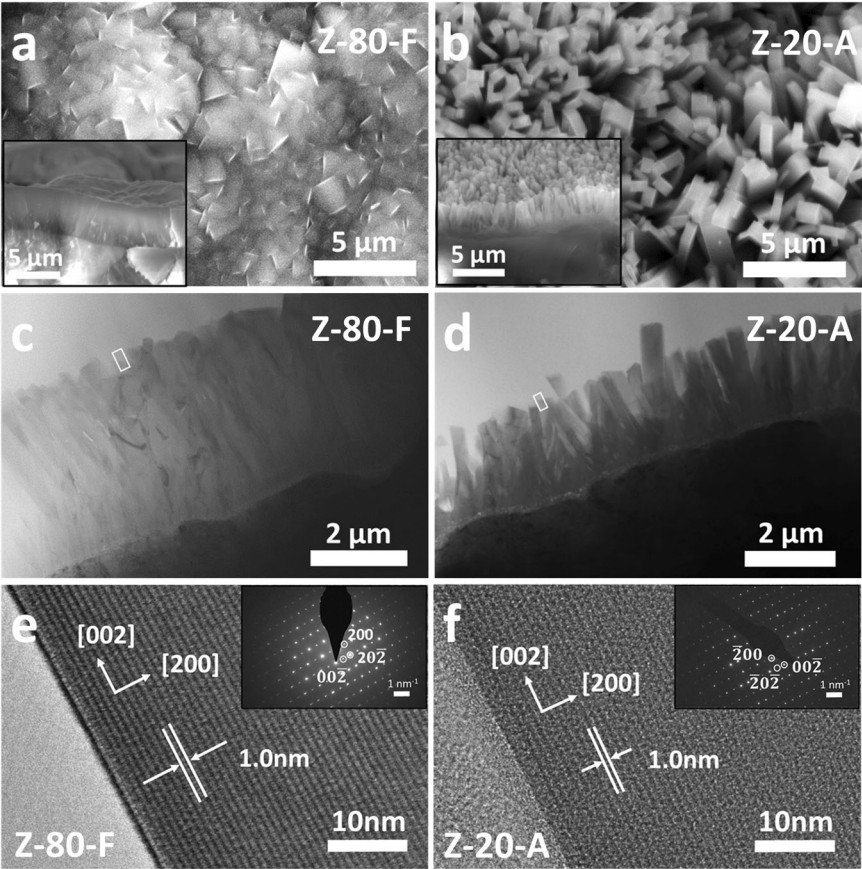

**Fig. 2 | Electron microscopy characterization of as-prepared Z-80-F film and Z-20-A arrays grown on cordierite monolithic substrates. a, b** Top view and cross-sectional view (inset) of SEM images. **c, d** Cross-sectional bright-field TEM images. **e, f** The high-resolution TEM images on the selected areas marked by rectangles in the (**c**) and (**d**) and corresponding selected area electron diffraction (SAED) patterns (inset).

**Table 1 | Quantitative summary of physicochemical characteristics of as-prepared conventional ZSM-5 film (Z-80-F) and array-structured ZSM-5 film (Z-20-A) and the propene adsorption performance of proton-exchanged samples (HZ-80-F and HZ-20-A)**

| Characteristics | Unit | Z-80-F | Z-20-A |
|---|---|---|---|
| $S_{BET, Sample}$[a] | $m^2\ g^{-1}$ | 84 | 66 |
| $S_{BET, ZSM-5}$[b] | $m^2\ g^{-1}$ | 323 | 338 |
| $S_{BET, Cordierite}$[b] | $m^2\ g^{-1}$ | 0.004 | 0.004 |
| $S_{Ext, ZSM-5}$[b] | $m^2\ g^{-1}$ | 100 | 145 |
| $V_{total, Sample}$[a] | $cm^3\ g^{-1}$ | 0.042 | 0.043 |
| $V_{meso, Sample}$[a] | $cm^3\ g^{-1}$ | 0.019 | 0.028 |
| Weak Acid Sites[c] | $mmol\ g_{HZSM-5}^{-1}$ | 0.22 | 3.00 |
| Strong Acid Sites[c] | $mmol\ g_{HZSM-5}^{-1}$ | 0.47 | 6.34 |
| $C_3H_6$ Adsorption Capacity[d] | $\mu mol\ mg_{HZSM-5}^{-1}$ | 0.92 | 1.45 |

[a]Cordierite substrates were included for the calculation of sample's surface area ($S_{BET,Sample}$), total pore volume ($V_{total,Sample}$), and mesopore volume ($V_{meso,Sample}$) in the $N_2$ adsorption-desorption isothermal test.

[b]The surface area of ZSM-5 film ($S_{BET, ZSM-5}$) was estimated by:

$S_{BET,Sample} \times Mass_{Sample} = S_{BET,Cordierite} \times Mass_{Cordierite} + S_{BET,ZSM-5} \times Mass_{ZSM-5}$.

Similarly, as cordierite substrate barely contributed to the sample's external surface area, assuming $S_{Ext,Cordierite} = 0$, the external surface area of ZSM-5 film ($S_{Ext, ZSM-5}$) could be calculated by:

$S_{Ext,Sample} \times Mass_{Sample} = S_{Ext,ZSM-5} \times Mass_{ZSM-5}$.

[c]The amounts of weak and strong acid sites were determined by ammonia temperature-programmed desorption.

[d]The propene adsorption capacity was calculated after 30 min of propene adsorption at 100 °C.

equally charged crystal surfaces are hindered. Accordingly, abundant Al ions in the solution may not only retard the ZSM-5 crystallization, leading to a lower loading in Z-20-A, but also prevent the crystals from intergrowing into each other during the synthesis process. As an alternative mechanism, the selective surface passivation through growth modifier addition may also be utilized to induce anisotropic crystal growth with preferential orientation, resulting in the array structured zeolite film. On the other hand, in this work, the synthesized ZSM-5 nanoarray shows a thick and uniform *b*-axis growth with no sign of growth suppression. Instead, a well-spaced crystal allocation can be achieved along the *b*-axis, ensuring the maximum exposure of *a* and *b* channels.

The unique array-structured morphology and lower Si/Al ratio will result in the distinction of Z-20-A in various physicochemical characteristics from Z-80-F. The $N_2$ adsorption–desorption isotherms displayed in Supplementary Fig. 6a and b demonstrate that, compared to Z-80-F, Z-20-A shows a lower $N_2$ adsorption uptake at low pressures ($P/P_O < 0.45$), a much larger area of hysteresis loop at relatively high pressure ($0.45 < P/P_O < 0.95$), and an enhanced uptake at high pressure ($P/P_O > 0.95$)[40]. When the contribution from cordierite support is excluded, the array-structured ZSM-5 film exhibits a higher specific surface area of 338 $m^2\ g^{-1}$ (Table 1). Especially, the external surface area increases from 100 $m^2\ g^{-1}$ in Z-80-F to 145 $m^2\ g^{-1}$ in Z-20-A. In addition, the results of the derived BJH pore size distribution reveal that Z-20-A exhibits more mesopores and macropores. As a result, despite the similar total pore volume in both samples, the mesopore volume in Z-20-A is ~47.4% higher than that in Z-80-F (Table 1).

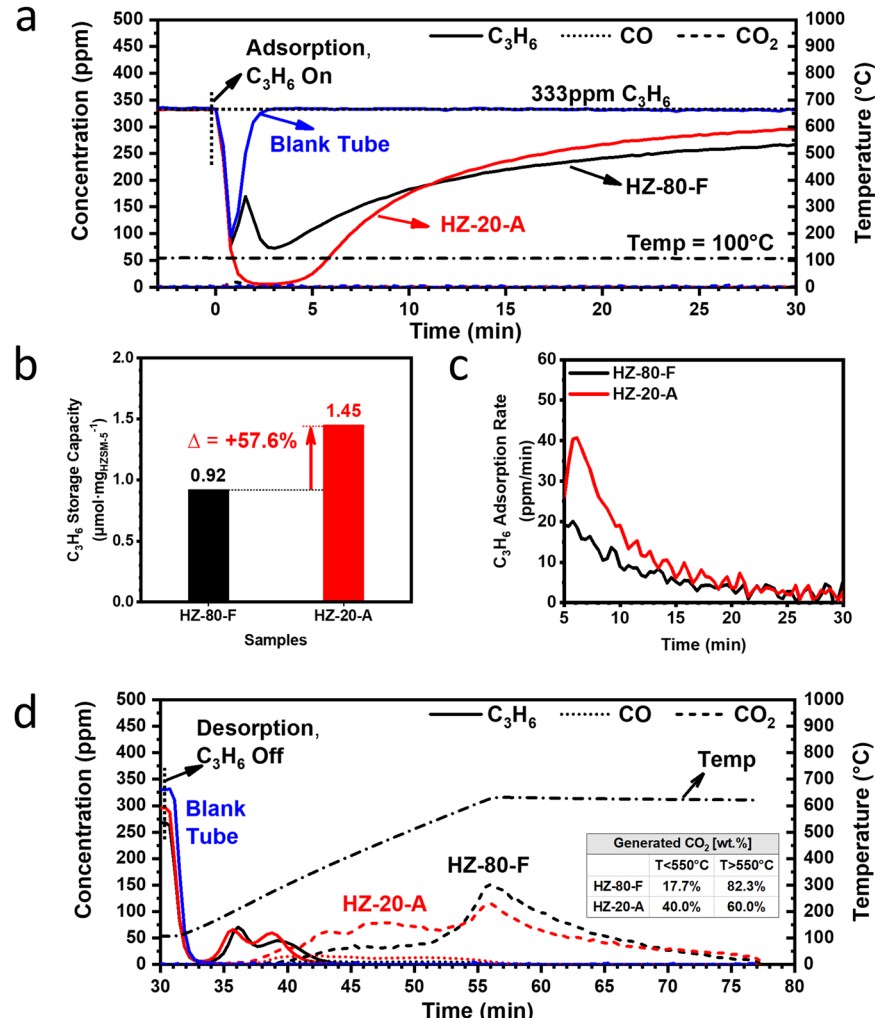

**Fig. 3 | Propene adsorption and desorption behavior of the proton-exchanged continuous ZSM-5 film (HZ-80-F) and array-structured ZSM-5 film (HZ-20-A).** **a** Propene adsorption behavior as a function of time. **b** Propene adsorption capacity. **c** Propene adsorption rate in the unit of ppm min$^{-1}$, as the first order derivative of $C_3H_6$ concentration change over test time. **d** Propene desorption behavior as a function of time. The percentage of the generated $CO_2$ at different temperature window during desorption on the two ZSM-5 samples are displayed in the inset.

Solid line: $C_3H_6$, dotted: CO, dashed: $CO_2$, and dashed dotted: Temperature (belongs to the 2nd $y$-axis). Black line: HZ-80-F, and red line: HZ-20-A. The result of blank tube testing in blue line is shown as reference. Each sample was maintained at 100 °C for 30 min under a gas mixture of 333 ppm $C_3H_6$ + 12% $O_2/N_2$, then heated to 600 °C at a ramp rate of 20 °C min$^{-1}$ under 12% $O_2/N_2$. Space velocity (S.V.): ~24,000 h$^{-1}$. Source data are provided as a Source Data file.

The presence of preferentially *c*-axis oriented nanorods in Z-20-A enables the pores and channels along *a*- and *b*-axes laterally parallel to the cordierite channel surface, facilitating the diffusion of reactant molecules. The higher specific surface area and mesopore volume in Z-20-A could further improve the diffusion and transportation of reactant molecules within the films and boost their reaction dynamics[24,41,42]. Conversely, the crystal intergrowth in Z-80-F may decrease the diffusion rates of reactants into and products out of the film, thus hindering the reactants' accessibility to the surface active sites. Therefore, the unique array-structured ZSM-5 film is potentially advantageous to mitigate the kinetic restrictions during the adsorption of gaseous molecules such as HCs.

The acid sites of HZ-80-F and HZ-20-A were examined by NH$_3$-TPD as shown in Supplementary Fig. 6c, d. Both samples exhibit two major peaks in the temperature regions of 150–225 °C and 300-400 °C, which could be ascribed to the chemisorption of NH$_3$ species on weak and strong acid sites, respectively. With the Si/Al ratio decreasing from 80 to 20, the amounts of both weak and strong acid sites increase remarkably. Accordingly, HZ-20-A possesses a much higher $C_3H_6$ adsorption capacity than HZ-80-F, because of its stronger adsorption affinity of unsaturated hydrocarbon due to a higher acidity. Meanwhile, the peaks of weak and strong acid sites shifted to a higher temperature from 184 °C and 326 °C in HZ-80-F to 207 °C and 360 °C in HZ-20-A, respectively.

## Performance benefits with improved reaction kinetics

Here propene ($C_3H_6$) adsorption at low temperature is used as a probe over proton-exchanged ZSM-5 monolithic samples (HZ-80-F and HZ-20-A), as evaluated in a homemade horizontal quartz tube reactor following the Low Temperature Storage Catalyst Test Protocol created by USDRIVE[43]. It is noted that $C_3H_6$ molecules with a kinetic diameter of ~4.95 Å can diffuse easily through the internal porosity of MFI structure[44]. The $C_3H_6$ adsorption capacity is related to the zeolite topology, specific surface area, Si/Al ratio, and the concentration and strength of acid sites[32,45,46]. Moreover, the adsorbed $C_3H_6$ molecules could further react and oligomerize, leading to the formation of short- and long-chain hydrocarbons and even coke[47,48].

As revealed in Fig. 3a, both HZ-80-F and HZ-20-A samples are not saturated by $C_3H_6$ after 30 min of adsorption at 100 °C. After being normalized by the mass of used HZSM-5, the specific $C_3H_6$ adsorption

capacity of HZ-20-A is 1.45 μmol $mg_{HZSM-5}^{-1}$, 57.6% higher than that of HZ-80-F (Fig. 3b and Table 1). The superior performance of HZ-20-A in $C_3H_6$ adsorption was further quantitatively demonstrated in a 120 min breakthrough test (Supplementary Fig. 7). Such an enhancement in adsorption capacity in HZ-20-A could be attributed to a larger amount of acid sites associated with its lower Si/Al ratio, as determined by the results of ammonia temperature-programmed desorption test (NH₃-TPD, Supplementary Fig. 6c, d and Table 1).

It is worth noting that HZ-20-A has faster $C_3H_6$ adsorption kinetics at 100 °C than HZ-80-F. On one hand, upon exposure, HZ-20-A quickly adsorbs $C_3H_6$, making the $C_3H_6$ concentration remains at a very low level for the first 5 minutes in the gas stream. In contrast, HZ-80-F shows a delayed initial adsorption and is not able to sufficiently remove $C_3H_6$, as a result of an obvious diffusion barrier in the dense continuous film. On the other hand, the cross point in the $C_3H_6$ concentration profiles of both samples in the following adsorption time ($\tau > 5$ min) implies a much faster $C_3H_6$ adsorption rate in HZ-20-A. Here the dynamic $C_3H_6$ adsorption rate is reflected by the first-order derivative of propene concentration change. Due to the above-mentioned nearly-completed propene removal on HZ-20-A and delayed initial propene adsorption on HZ-80-F at the first several minutes of adsorption, only the rates at $\tau > 5$ min are compared and displayed in Fig. 3c to demonstrate a faster propene adsorption rate in HZ-20-A than that in HZ-80-F throughout most of the time. Therefore, HZ-20-A is of fast adsorption kinetics as a promising HC trap device for the desired short cold-start period.

After the adsorption for 30 min, $C_3H_6$ is removed from the feed gas while the temperature starts to increase from 100 °C to 600 °C. As revealed in Fig. 3d, the adsorbed $C_3H_6$ is released from samples and oxidized in the temperature ramp stage, and the oligomerized species generated during adsorption will also be combusted. Here $CO_2$ is the major product while CO is negligible. With a larger amount of acid sites, HZ-20-A is supposed to contain more oligomerized products over HZ-80-F after 30 min adsorption, as supported by a higher weight loss in TGA tests on the adsorbed samples (see Supplementary Fig. 8). Correspondingly, HZ-20-A might generate more $CO_2$ than HZ-80-F throughout the temperature ramping stage. However, the test results show that HZ-20-A yields less $CO_2$ and a lower percentage of total generated $CO_2$ above 550 °C than HZ-80-F. Considering the generated $CO_2$ at a higher temperature is due to the combustion of long-chain HCs and coke, a less intensified oligomerization is suggested in HZ-20-A than HZ-80-F. This might point to the enhanced diffusion of reactants in and reaction products out of the ZSM-5 nanoarrays enabled by c-axis oriented ZSM-5 nanorod crystals and unique array-structured morphology[30,41]. Thus, such a ZSM-5 array structure can help mitigate the long-chain HCs and coking formation during a chemical process in general.

In situ diffuse reflectance infrared Fourier transform spectroscopy (DRIFTS) was carried out to investigate the surface molecular species on HZ-80-F and HZ-20-A samples during the $C_3H_6$ adsorption and desorption process (see Supplementary Figs. 9 and 10). Due to the co-presence of ZSM-5 and monolithic substrates under propene sorption process, it is intrinsically challenging to assign the observed bands in the full DRIFTS spectra. Thus, only selected bands are presented in the following discussions. At 100 °C, a set of bands can be observed that belong to propene itself and derived oligomers due to adsorption. The peak at 1470 $cm^{-1}$ could be assigned to the vibration of H-$C_3H_6$ adsorbed via their π bonds[49]. In addition, at the high-IR range from 2700 to 3100 $cm^{-1}$, $-CH_3$ symmetric stretching (2933 $cm^{-1}$), asymmetric stretching (2960 $cm^{-1}$), and $-CH_2-$ stretching (2868 $cm^{-1}$) modes of oligomeric species can be observed[50]. It is noted that the peak areas of H-$C_3H_6$ and $CH_2$-/$CH_3$- in HZ-80-F and HZ-20-A increase gradually with adsorption time. By integrating the peak areas of H-$C_3H_6$ (1470 $cm^{-1}$) and $CH_2$-/$CH_3$- (3080–2750 $cm^{-1}$) and setting the integrated values at 30 min as the reference, we could roughly quantify

the increasing rate of each species in both samples. As such, the faster rates of both $C_3H_6$ adsorption and oligomerization are proved in HZ-20-A. For instance, at $\tau = 4$ min in Supplementary Fig. 9, the relative integrated areas of H-$C_3H_6$ and $CH_2$-/$CH_3$- in HZ-20-A are ~2.5 and ~1.5 times of those in HZ-80-F, respectively. Following a similar approach, faster rates of H-$C_3H_6$ desorption and decomposition of alkoxide moieties in HZ-20-A during the temperature ramp when referred to the peak area at $T = 100$ °C were also demonstrated in Supplementary Fig. 10.

As mentioned earlier, the acid sites in H-ZSM-5 are critical in $C_3H_6$ adsorption, playing roles as both adsorption centers and catalytic active centers for oligomerization[13,51]. The enrichment of captured $C_3H_6$ could promote the level of catalytic oligomerization reactions, which was also beneficial to the adsorption process. However, the oligomerization products may negatively impact both reactions if they cannot timely diffuse out due to a potential diffusion barrier from zeolite micropores and block the accessibility to the acid sites. This phenomenon was observed in the adsorption behavior of washcoated HZP-26 and HZP-87 samples (Supplementary Fig. 11). Given the similar amount of washcoat loading and intrinsic structure properties of the commercial powders in crystal size and crystallization level, HZP-26, even with more acid sites than HZP-80, showed a lower $C_3H_6$ adsorption capacity. The texture characteristics of washcoated HZP-87 and HZP-26 are shown in Supplementary Table 2. HZP-87 has a higher external surface area while the total pore volume is similar, leading to a potentially more accessible adsorption site for propene guest molecules in HZP-87. In addition, the oligomerization products may block the adsorption sites for propene if they could not diffuse out timely, which is also affected by the texture properties of washcoated samples. The opposite trend of $C_3H_6$ adsorption capacity in HZ-20-A vs. HZ-80-F, and HZP-26 vs. HZP-80 strongly evidenced the advantages of the array structure, where both reactant-in and product-out processes could be facilitated, resulting in less residual oligomerized products in HZ-20-A during adsorption.

Nanoarray structured catalysts have been reported with superior robustness in mechanical stability and performance reproducibility over the conventional washcoated catalysts[34]. Supplementary Fig. 12 displayed the negligible weight loss of the ZSM-5 nanoarray samples after 30 min sonication, in a sharp contrast with a significant 16% loss of the washcoated samples even with α-$Al_2O_3$ binder. Furthermore, the results of $C_3H_6$ adsorption testing in Supplementary Fig. 13 and calculated $C_3H_6$ adsorption capacity in Supplementary Table 3 from either the batch-to-batch or test-to-test investigation demonstrated the strong performance reliability of ZSM-5 nanoarray.

Finally, to validate the advantages of HZ-20-A design in HCs removal, especially for a short duration at low temperature, propene gas mixture was passed through a sequentially arranged dual-bed reactor system including a propene trap (i.e., HZ-80-F or HZ-20-A) and a propene oxidation catalyst (i.e., $Co_3O_4$ nanoarray, Supplementary Fig. 14) following the cold-start test (CST) protocol by USDRIVE[43]. The reactor was maintained at 100 °C for 3 min and then heated to 600 °C at a ramp rate of 20 °C $min^{-1}$. From the CST results as illustrated in Fig. 4, $Co_3O_4$ nanoarray could catalytically oxidize $C_3H_6$ into $CO_2$ starting from ~200 °C[52], but its propene adsorption capability is negligible at lower temperatures. Instead, the presence of either HZ-80-F or HZ-20-A in front of $Co_3O_4$ could significantly improve the $C_3H_6$ removal efficiency at $T < 200$ °C when $Co_3O_4$ is not thermally activated, although the $C_3H_6$ oxidation performance to $CO_2$ at higher temperature is compromised slightly. No CO is detected throughout all the tests. To be noted, compared to HZ-80-F + $Co_3O_4$, HZ-20-A + $Co_3O_4$ could completely remove $C_3H_6$ at 100 °C for the dwelling time of 3 min, and exhibit a similar T90 temperature where 90% of $C_3H_6$ are completely oxidized to that when only $Co_3O_4$ is used at the oxidation stage. Thus, the nanorod array-structured design in ZSM-5 adsorber offers a promising route to reduce hydrocarbon emission during the

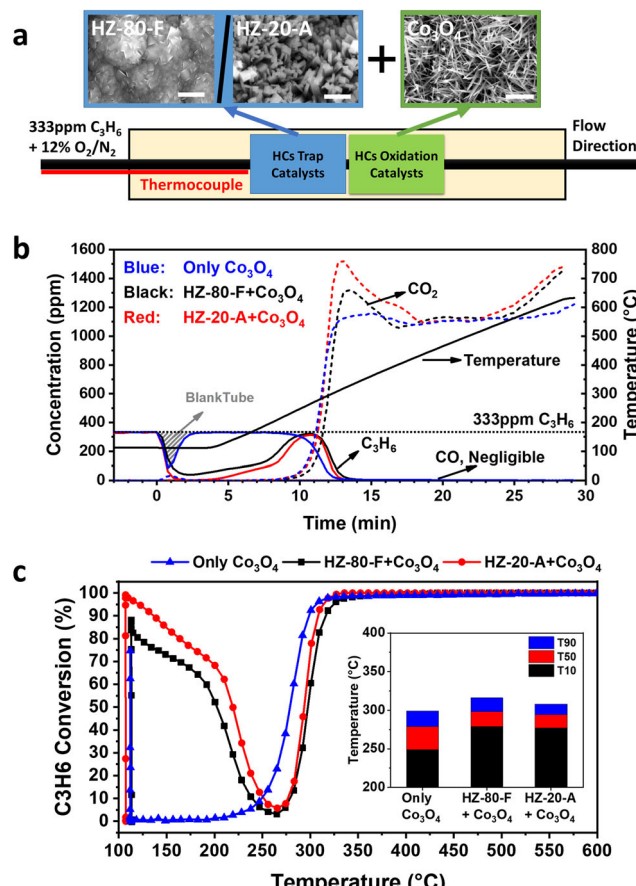

**Fig. 4 | Performance of cold-start test using a sequentially placed dual-bed reactor of as-synthesized ZSM-5 samples and Co$_3$O$_4$ nanoarray in sequence. a** Schematic of the designed dual-bed reactor system for a cold-start test (CST) including a propene trap in front (blank, HZ-80-F, or HZ-20-A) and a propene oxidation catalyst at the downstream (Co$_3$O$_4$ nanoarray). Top insets: corresponding SEM images of HZ-80-F, HZ-20-A and Co$_3$O$_4$ nanoarray in a scale bar of 5 μm. **b** Cold-start test (CST) results over the test time. The gray shaded area reflects the performance of blank tube testing. **c** Propene conversion efficiency vs. reaction temperature for different configurations. The inset shows the temperature of T10, T50 and T90 where 10%, 50% and 90% of propene conversion efficiency is achieved. Solid: C$_3$H$_6$, dotted: CO (negligible), dashed: CO$_2$. Blue line: only Co$_3$O$_4$, black line: HZ-80-F + Co$_3$O$_4$, and red line: HZ-20-A + Co$_3$O$_4$. CST was carried out under a gas mixture of 333 ppm C$_3$H$_6$ + 12% O$_2$/N$_2$. Each sample was kept at 100 °C for 3 min and then heated to 600 °C by 20 °C min$^{-1}$. Space velocity (S.V.): ~24,000 h$^{-1}$. Source data are provided as a Source Data file.

short cold-start period at a low temperature while retaining the downstream oxidation catalyst activity.

ZSM-5, as a versatile and abundant material, has uniform subnanometer-sized pore channels, tunable composition and acidic properties, and tailorable morphology. It is widely used in the chemical and petrochemical industries as selective catalysts, adsorbents, and membranes. Depending on the applications and target reactants, e.g., selective adsorption and separation for gases, catalytic reforming (methanol to gasoline, fluid catalytic cracking, and alkylation) or filtration (heavy metal and dye removal) etc., manipulating the ZSM-5 structures is vital to achieve a desirable performance. This is especially true considering the anisotropic pore distribution of ZSM-5. The anisotropic channel of ZSM-5 plays an essential role in determining the catalytic properties besides the acid properties and micropore size and geometry[53]. ZSM-5 contains two types of interconnected 10-membered ring channels: the straight channel along the *b*-axis (5.6 × 5.3 Å) and the sinusoid one alone the *a*-axis (5.5 × 5.1 Å). NMR studies[54] have shown that diffusion in the direction of the *a*-axis of the crystals is much

slower than in the *b*-axis direction. Typically, decreasing the *b*-axis channel length could shorten the path length of molecular diffusion and enhance accessibility to internal acid sites, which could enhance the activity of the catalyst in the acid-catalyzed reaction and reduce deactivation. Therefore, most of the literature reported work focuses on controlling ZSM-5 *b*-axis oriented growth to achieve a very thin thickness along *b*-axis, ranging from serval nanometers to hundreds of nanometers[53–56]. Guest molecules such as methanol, benzene, n-heptane, and phenol etc. for alkylation and methanol-to-other catalytic reactions have been used to showcase the superior catalyst lifetime and activity. While limited work is available in terms of the *a*- and *c*-axes orientation control. Therefore, current work fills the gap and points out a possible route to push the ZSM-5 catalytic applications toward fast kinetics and high capacity. It is noted that the orientation control strategy is seldomly utilized for gas separation applications. Instead, tuning of the Al/Si ratio[57], pore modification[58] and ion exchange[59,60], etc. approaches, based on a membrane configuration, are typically applied to achieve improved gas permeability and selectivity. Considering the smaller zig-zag channels along *a*-axis which can provide extended zeolite-guest molecules interaction, controlling of *a*-axis growth could offer a promising pathway to achieve improved performance for gas separation.

In summary, *c*-axis oriented array-structured ZSM-5 film was successfully synthesized on the cordierite honeycombs with the assistance of precoated silicalite-1 seeds. With a decreased Si/Al ratio and adjusted alkalinity, ZSM-5 film evolves from a conventional dense film composed of highly-intergrown crystals to a nanorod array-structured morphology with preserved single crystalline nanorod individually. The performance results and in-situ DRIFTS studies have demonstrated that, despite a lower loading due to the higher Al content, the array-structured ZSM-5 film exhibits much higher C$_3$H$_6$ adsorption capacity and faster adsorption kinetics at low temperature, which is attributed to the abundant acid sites and large specific surface area and mesopore volume. In addition, the oriented array-structured ZSM-5 film on monolith also helps mitigate the long-chain HCs and coking formation due to the enhanced diffusion of reactants in and reaction products out of the array structures. Furthermore, the combination of as-developed array-structured ZSM-5 film with downstream oxidation catalysts could achieve multifunctional hydrocarbon adsorption and oxidation, efficiently reducing the hydrocarbon emission throughout a wider temperature window.

It is worthing noting that to reveal exact structure-function relationships, some important questions remain open on the heterogeneous adsorbers and catalysts with ZSM-5 nanorod arrays demonstrated here. For example, further in-situ/Operando structure characterizations and characterization technique development are needed for decoupling the sorption and reaction functions of individual constituents in such a multi-component structured device during operation. These new characterization efforts will help unravel the role monolithic substrate plays on the sorption behavior of ZSM-5 nanorod array complementing to the in situ DRIFTS study in this work. In addition, refining synthesis control of array-structured ZSM-5 film on monoliths and understanding its exact growth mechanism can help identify and understand the structure (e.g., crystallite size, intercrystallite spacing) and chemical composition (e.g., Si/Al ratio) effects on the sorption and reaction kinetic performance.

Nevertheless, the array-structured film design could offer energy-efficient solutions to overcome both sorption and reaction kinetic restrictions of small gaseous molecules in various solid porous materials such as MOF and zeolitic materials that possess anisotropic surfaces, microporosities, and channels. Besides, the enhanced sorption kinetics and capacity through array structured zeolites could be broadly implied in various chemical and physical processes where kinetics and capacities are the key for improved process technologies, such as carbon capture and conversion, clean energy and chemical

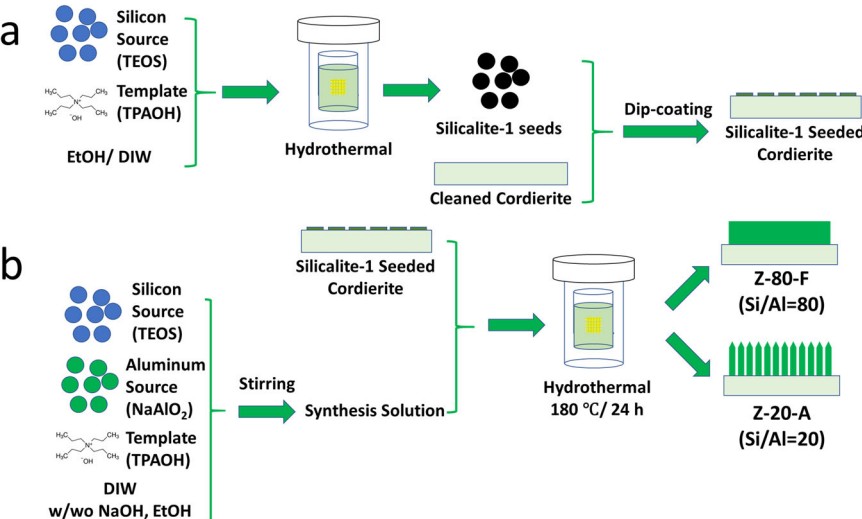

**Fig. 5 | Schematics of the synthesis process of ZSM-5 films on the cordierite substrate by secondary-growth. a** A seeding process on bare cordierite substrate by dip-coating self-prepared silicalite-1 seeds; **b** Growth of ZSM-5 films at different Si/Al ratios of 80 and 20 on the seeded substrate.

transformation, environmental remediation, related greenhouse gas emission abatement, and others.

## Methods

### Materials

Cordierite (400 cpsi, Corning Inc.) was used as substrate for the sample preparation.

To synthesize ZSM-5 films, tetrapropylammonium hydroxide solution (TPAOH, 25 wt.% in water, ACROS), tetraethyl orthosilicate (TEOS, 98 wt.%, ACROS), sodium aluminate (NaAlO$_2$, 92 wt.%, Fisher Scientific), and sodium hydroxide (NaOH, pellets, ACROS) were purchased. Ammonium nitrate (NH$_4$NO$_3$, pallets, Fisher Scientific) was used for the proton ion exchange purpose.

Cobalt chloride hexahydrate (CoCl$_2$·6H$_2$O, ACROS) and urea ((NH$_2$)$_2$CO, Fisher Scientific) were used for the synthesis of Co$_3$O$_4$ nanoarray.

### Catalysts preparation

**Synthesis of ZSM-5 films at different Si/Al ratios.** ZSM-5 films at different Si/Al ratios (SA, 20 or 80) were hydrothermally synthesized on the cordierite substrate by secondary growth method. The synthesis process is shown in Fig. 5, including the preparation of silicalite-1 seeds, deposition of silicalite-1 seeds on cordierite surface by dip coating, and the following hydrothermal synthesis of ZSM-5 films.

**Preparation of silicalite-1 seeds.** The silicalite-1 seeds were prepared using the following procedure. Firstly, 25.8 g TPAOH and 21.5 g ethanol (EtOH) were mixed under magnetic stirring at a rate of 500 rpm. Then 24.3 g TEOS was gradually added by droplets into the pre-mixed solution. After mixing at 30 °C for 6 h, the as-prepared solution was transferred into a Teflon-lined stainless-steel autoclave for the reaction at 100 °C for 72 h. The final obtained solution was stored in a refrigerator for the seeding purpose.

**Deposition of silicalite-1 seeds by dip coating.** The obtained silicalite-1 seeds solution was diluted by 10 times with 430 H$_2$O/100 EtOH (vol.%) for the seeding purpose. A piece of pre-cleaned cordierite substrate in a dimension of 2 cm × 2 cm × 1 cm was submerged into the diluted seeding solution while being sonicated for 30 s in a sonicator (Branson 5510, 42 kHz, 135 W), and blown by the compressed air to remove the extra residual solution. Next, the substrate was dried under microwave irradiation (Sharp R-309yw, 1000 W) for 80 s, and treated in an oven at 350 °C for 5 min to stabilize the seeds deposition. The

above dip-coating steps were repeated for 3 times to ensure that the substrate surface was fully covered by a layer of silicalite-1 seeds. Finally, the seeded substrate was calcined at 500 °C for 1 h in a furnace at a ramp rate of 5 °C min$^{-1}$.

**Synthesis of ZSM-5 films on the cordierite.** ZSM-5 films at different Si/Al ratios of 20 and 80 were hydrothermally synthesized on the seeded substrate. In the case of ZSM-5 film at Si/Al = 20, a solution mixture with a molar ratio of 1TEOS: 0.112TPAOH: 0.05NaAlO$_2$: 111H$_2$O: 0.36NaOH: 8EtOH was prepared. 40 mL of the mixture solution was then transferred into a 50 mL Teflon-lined stainless-steel autoclave, where a piece of pre-seededcordierite substrate was vertically submerged and suspended in the solution. After the reaction at 180 °C for 24 h, the autoclave was quenched by the flowing water to room temperature. Then the sample was withdrawn and sonicated in DI water for 30 min to remove any loosely sedimented crystals. The sample was dried at 110 °C overnight and calcined at 550 °C for 4 h to burn out the occluded template. Following similar procedures, ZSM-5 film at Si/Al = 80 was prepared using a solution of 1TEOS: 0.112TPAOH: 0.125NaAlO$_2$: 111H$_2$O. The as-prepared ZSM-5 films at different Si/Al ratios of 20 and 80 were denoted as "Z-20-A" and "Z-80-F", respectively. In addition, to investigate the growth process of different films, samples synthesized at 3, 6, 12, 18, and 24 h were also prepared and marked as "X_Y", where X represented "Z-20-A" or "Z-80-F", and Y represented the synthesis hours.

The as-prepared ZSM-5 films were ion-exchanged with 1 M NH$_4$NO$_3$ aqueous solution at 80 °C for 12 h. After that, each sample was dried at 110 °C overnight. The ion-exchanged procedures were repeated by 3 times before the final calcination at 500 °C for 4 h (2 °C min$^{-1}$). The obtained proton-exchanged ZSM-5 films were denoted as "HZ-20-A" and "HZ-80-F", respectively.

### Synthesis of washcoated ZSM-5 samples

Two commercial H-ZSM-5 powders at Si/Al ratios of ~26 and ~87 were purchased from ASC Material. Based on the information provided by the manufacturer, both H-ZSM-5 powders had similar intrinsic properties, including a crystal size of 300 nm, BET surface area of 362 m$^2$ g$^{-1}$, and 100% relative crystallization. The commercial H-ZSM-5 powders, α-Al$_2$O$_3$ binder, and DI water (19:1:80, wt.%) were mixed as a slurry for the following washcoat. During the washcoat process, a clean bare substrate was dip-coated by the slurry and dried under microwave assistance for multiple cycles to achieve a similar desired loading amount of H-ZSM-5, saying ~23.2 wt.% for Si/Al of ~26 and ~22.4 wt.% for Si/Al of ~87. Finally, the washcoated samples were calcined at 550 °C for 4 h at a

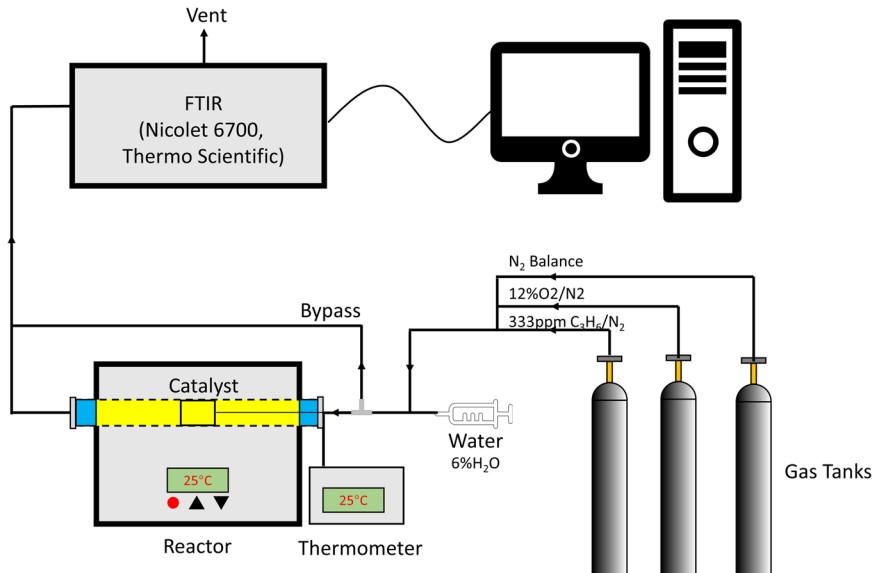

**Fig. 6 | Schematic of the reactor setup for propene adsorption and cold-start test.** The test setup includes a microreactor with temperature-programmed control, gas line with mass flow control, an FTIR spectrometer, and a computer for data recording.

ramping rate of 2 °C/min. The samples were marked as HZP-26 and HZP-87, respectively, based on the Si/Al ratio of each commercial H-ZSM-5 powder.

### Synthesis of Co₃O₄ nanoarray-based monolithic catalysts

$Co_3O_4$ nanoarray were in situ grown on the cordierite surface via a modified solution chemistry strategy. Briefly, 10 mmol $CoCl_2$ as cobalt precursors and 60 mmol urea were dissolved in 30 ml of distilled water and ultra-sonicated to achieve a clean solution. A piece of pre-cleaned cordierite was suspended into the as-prepared solution for hydrothermal synthesis. The reaction was held at 98 °C for 6 h. After that, the cordierite substrate was rinsed with distilled water and ethanol for several times, dried overnight, and calcined at 350 °C for 2 h at a ramp rate of 5 °C min⁻¹. The loaded amount of $Co_3O_4$ nanoarray is ~10 wt.% of the cordierite mass.

### Catalysts characterization

The morphology and structure of the ZSM-5 films were investigated by a filed-emission scanning electron microscope (SEM, Teneo LVSEM, FEI) at an accelerating voltage of 20 kV and a high-resolution scanning transmission electron microscope (STEM, Talos, FEI; 200 kV) combined with energy dispersive spectroscopy (EDS). The cross-sectional (S)TEM samples were characterized with a Tecnai 30 STEM at an accelerating voltage of 300 kV.

X-ray diffraction (XRD) patterns of the as-prepared ZSM-5 films were acquired on a BRUKER D2 X-ray diffractometer using Cu-Kα radiation (1.5406 Å) operated at 30 kV and 10 mA. The scan range was from 5 ° to 35 ° at a scan rate of 2.4 ° min⁻¹. The value of crystallographic preferred orientation (CPO) is calculated to evaluate the tendency of the film orientation.

The $N_2$ adsorption-desorption isotherms were measured at 77 K on a Micromeritics ASAP 2020 volumetric adsorption analyzer to characterize the specific surface area (BET, Brunauer-Emmett-Teller plot) and pore size distribution (BJH, Barrett-Joyner-Halenda model). Each sample was degassed under vacuum at 150 °C for 6 h before the measurement to remove water and other possible adsorbed species.

To determine the acid sites on each sample, ammonia temperature-programmed desorption ($NH_3$-TPD) was carried out in a homemade horizontal quartz tube reactor. Each sample was pre-treated in $N_2$ flow (200 sccm) at 550 °C for 1 h and then cooled down to 100 °C. Then 1% $NH_3$ in $N_2$ (50 sccm) was fed in to saturate the sample for 1 h. The physisorbed $NH_3$ was blown away by purging the sample in

pure $N_2$ flow (200 sccm) for 1 h. Subsequently, the sample was heated in $N_2$ flow from 100 °C to 500 °C at a ramp rate of 10 °C min⁻¹. The desorbed $NH_3$ was monitored by a Fourier Transform infrared spectrometer (FTIR, Nicolet 6700, Thermo Scientific).

In situ diffuse reflectance infrared Fourier transform spectroscopy (DRIFTS) was conducted on a Fourier Transform Infrared Spectrophotometer (FTIR, Nicolet iS-50, Thermo Scientific) equipped with an MCT detector cooled by liquid nitrogen. Specifically, the ZSM-5 film-based monoliths (3×3 channels) were crushed into powders and ground. The sample was pretreated in 12% $O_2/N_2$ at 500 °C for 1 h to eliminate any potential contaminants and cooled down to 100 °C for background spectrum collection. While the spectrum was continuously detected at a resolution of 2 cm⁻¹ for 32 scans, the sample was firstly interacted with 333 ppm $C_3H_6$ + 12% $O_2/N_2$ at a flow rate of 50 ml min⁻¹ at 100 °C for 30 min, followed by purging with 12% $O_2/N_2$ for another 30 min, and finally heated to 450 °C at a rate of 2 °C min⁻¹ with 12% $O_2/N_2$.

### Performance evaluation of propene adsorption

Propene adsorption performance of the as-prepared two ZSM-5 films was evaluated following the Low Temperature Storage Catalyst Test Protocol created by USDRIVE in the same setup for $NH_3$-TPD, as shown in Fig. 6. Each sample with a volume of 0.7 cm × 0.7 cm × 1 cm was wrapped with a catalyst cushion mat and then loaded into the center of quartz tube (1 in. diameter). The cushion mat will be thermally expanded after high-temperature exposure to fill up the gaps inside the quartz tube and force the whole gas flow to only pass through the monolithic catalysts. After being calibrated for $C_3H_6$, CO, and $CO_2$, FTIR (Nicolet 6700, Thermo Scientific; equipped with 2 Meter Gas Cell in 200 ml internal cell volume) was used to detect the concentration of inlet and outlet gases. FTIR was set with number of scans of 16, resolution of 0.5, data spacing of 0.241 cm⁻¹, and estimated sampling interval of 26 s. Before each test, the sample was pretreated at 600 °C for 20 min under the atmosphere of 12 vol.% $O_2$ + 6 vol.% $CO_2$ + 6 vol.% $H_2O$ in $N_2$ balance. After the reactor was cooled down to 100 °C in $N_2$ flow, background FTIR spectrum was collected for the following continuous detection. The feed gas with a composition of 333 ppm $C_3H_6$ + 12 vol.% $O_2/N_2$ firstly bypassed the reactor for 3 min to ensure a stable concentration and then switched back to the reactor while the temperature was maintained at 100 °C. After 30 min, $C_3H_6$ was cut off from the feed gas while the reactor was heated to 600 °C with a ramp rate of 20 °C min⁻¹. The space velocity was set to ~24,000 h⁻¹.

## Cold-start test

The cold-start tests (CST) on the dual-bed reactor system using HZ-80-F or HZ-20-A adsorber and $Co_3O_4$ nanoarray oxidation catalyst were performed using the same experiment setup and FTIR in $NH_3$-TPD and propene adsorption testing. All samples had the same volume of 0.7 cm × 0.7 cm × 1 cm. The test procedures were similar to the performance evaluation of propene adsorption, except that the adsorption stage at 100 °C was maintained for only 3 min before ramping the temperature to 600 °C at 20 °C/min, and the gas composition was remained with 333 ppm $C_3H_6$ + 12 vol.% $O_2/N_2$ throughout the test. The space velocity was set to be ~24,000 $h^{-1}$.

## Data availability

The data that supports the findings of the study are included in the main text and supplementary information files. Source data are provided in the Source Data file. Additional raw data can be obtained from the corresponding author upon request.

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

## Acknowledgements

We acknowledge the financial support from the US National Science Foundation (Award Nos. IIP1919231 and CBET1344792), and the US Department of Energy (Award No. DE-EE0006854). The authors are also grateful for the support from the University of Connecticut CARIC program. J.W. was partially supported by a Thermo Fisher Scientific Graduate Fellowship. The microscopy studies were partially performed using the facilities at the UConn/Thermo Fisher Scientific Center for Advanced Microscopy and Materials Analysis (CAMMA).

## Author contributions

P.-X. G. conceived the research; P.-X. G. and J. W. designed the experiments and co-wrote the manuscript; J. W. synthesized and characterized the samples; J. W. performed most of the catalyst evaluation, data collection, and analyses; C. Z. and B. Z. conducted the bench reactor testing and in situ DRIFTS; W. T. and X. L. performed temperature programmed testing and physico-chemical characterization; M. W. helped with the probe catalytic reaction testing; F. L. and Y. D. performed the high-resolution TEM characterization; and all the authors discussed the results and provided input on the manuscript.

## Competing interests

A US patent application on the method of making ZSM-5 array structures, with P.-X. G., J. W., C. Z., and B. Z. as the co-inventors, has been filed through the University of Connecticut[61]. The remaining authors declare no competing interests.
