## [Peer Review File · Nature Communications]

Enhancing Sorption Kinetics by Oriented and Single Crystalline Array-structured ZSM-5 Film on MonolithsEditorial Note: Parts of this Peer Review File have been redacted as indicated to remove third-party material where no permission to publish could be obtained.

REVIEWER COMMENTS

Reviewer #1 (Remarks to the Author):

The manuscript discusses an innovative approach to enhance the adsorption and desorption properties of zeolites for cold trap applications. The process of adsorbing hydrocarbons at low temperatures and releasing them at higher temperatures is crucial for vehicle emission control, especially considering the low activity of catalysts during cold start periods.

The concept presented in the manuscript is both intriguing and novel. Through the manipulation of the orientation of zeolite (MFI) crystals, the authors achieved an enhanced adsorption rate. This adjustment in adsorption kinetics played a vital role in minimizing oligomer formation, showcasing the effectiveness of the proposed method.

Notably, the authors conducted tests under conditions closely resembling real-world applications, a rarity in research focused on the development of cold trap adsorbents. Consequently, I recommend the publication of this work, contingent upon the authors addressing the following technical concerns:

1. The manuscript lacks clarity on how the authors controlled the orientation of zeolite crystals. Although the synthesis methods indicate variations in Si/Al ratios and NaOH concentrations in the two zeolite samples, the authors should provide insights into the control of orientation.
2. Given the significant roles of Si/Al ratios in oligomer formation and desorption temperature, I suggest exploring the creation of ZSM-5 with Si/Al=80 and a c-oriented array. This would enable an independent study of the effects stemming from both Si/Al ratio and orientation.
3. Considering the desorption temperature exceeding 500°C for complete oligomer desorption, there is a concern about its practicality for cold trap applications. The authors should address whether this temperature is too high for such applications.

Reviewer #2 (Remarks to the Author):

In this paper, the author wished to demonstrate the superior adsorptive and catalytic performances of a ZSM-5 film specifically designed with nanocrystals oriented perpendicular to the substrate as compared with a more conventional film with a crystal orientation parallel to the surface. It is proposed that the peculiar array structuration of the nanocrystals both enhances the sorption kinetics and sorption capacity of propene while decreasing coking effects during desorption. The studied context is that of cold-start application where such requirements are important.

Although this research is important for the adsorption and catalysis community and was rather well carried out in many aspects, some issues remained to be addressed before to consider the manuscript for publication. Overall, I am not fully convinced of the presented performances and related explanations even if it clear that nanostructuration may be helpful in achieving better performances.

It seems difficult to compare the performances from samples having a different Si/Al ratio because the latter parameter is very influent on propene adsorption, especially . Other problems are related to the testing methodologies (breakthrough tests and DRIFTS). The DRIFTS section is not useful because the data seem not very reliable and not interpreted correctly.

Specific comments are listed below :

General :

- All along the manuscript, there are many problems related to the insertion of links to the figures in the pdf file (error - reference source not found).

- Is there any specific reason for the choice of ZSM-5 as preferred zeolite for the trapping of propene ? Although this zeolite was demonstrated to be interesting for cold-start application, its porosity is made from interconnecting straight and zigzag channels. Hence, it could possibly be more complex to address the effect of pore orientation as compared with pseudo-1D zeolites such as mordenite, which is also important for cold-start as shown for instance in refs. 28 and 29.

Catalyst preparation and characterization :

- Although interesting, the effect of the Si/Al ratio on the film structuration is not very clear. At the bottom of page 4, the authors suggest that the higher concentration of Al during synthesis of Z-20-A (as compared with Z-80-F) is responsible of the differences observed, presumably through a repulsion mechanism in the initial stages of crystal growth. In that respect, it would be important to examine other Si/Al ratio to confirm this.

Alternatively, it is also suggested in the general literature that anisotropic growth could be due to some adsorption of some specific species of the sides of rods, preventing their growth on corresponding directions.

Please comment on these points.

- The N₂ adsorption isotherm of cordierite substrate has to be given and its porous characteristics have to be included in Table 1.

- Where are presumably located the mesopores in both films ? They are not visible in Fig. 1. Could the improvement suggested by the authors on adsorption kinetics could be related to the presence of such mesopores rather than a different orientation of straight and zigzag channels ?

Performance Benefits Demonstration: Improved Reaction Kinetics

- The amount and type of acid sites on both zeolite films, as measured from ammonia adsorption, is not commented in the text. However, it is about more than 10 times higher for Z-20-A as compared with Z-80-F, while the acid strength is not very different. Considering that acid sites are thought to be responsible for propene adsorption (as also shown in provided reference 28), it may be expected that the propene adsorption capacity may change by the same order of magnitude. However, it is not the case (Table 1). How do the authors explain this ?

- I find that the shape of breakthrough curves in Fig. 2 and Fig. S7 is a bit odd, as a period of total retention of propene is never reached for Z-80-F. Hence, i'm not sure that the methodology used to record the data is accurate enough. For dynamic adsorption tests, how are the samples introduced in the tube to ensure that the whole flow is passing through and not on the sides ? What is the sampling interval to measure propene concentration all along the breakthrough ? Why using a temperature of 100°C instead of a lower temperature to ensure that adsorption processes are not occurring together with catalytic ones.

- The points could be shown on breakthrough curves and more details on the experimental methodology have to be added in the supplementary section. A scheme of the adsorption setup has to be given likewise some details about FTIR (number of scans, sampling interval, cell volume etc....).

- Considering these last remarks, I am not very convinced about the discussion on adsorption kinetics given on the top of page 7 likewise that on desorption behaviour on the same page. In my opinion, the desorption/oxidation profiles are very similar for both zeolites.

- DRIFTS page 10. The assignment of infrared bands has to be done with much more accuracy, starting from propene in the gas phase and then coming back to adsorbed species. What are the bands located at 1300 and 1175 cm^{-1} ? What about the C=C stretch located around 1650 cm^{-1} and the evolution of acid O-H groups ? Note that the 1470 cm^{-1} is one of C-H bending modes.

- Moreover, the profiles of DRIFT spectra are influenced by the zeolite absorption, which is very strong below 1600 cm^{-1} . From the reviewer experience, the bands of adsorbed species could be very much perturbed and it is not possible to do quantitative measurements by DRIFTS. Transmission with pellets if of greater help here. For comparison and better interpretation, the authors should pay attention to J. Phys. Chem. C 2018, 122, 11, 6128–6136

- I don't understand the trends observed for the washcoated samples in function of Si/Al ratio. Higher propene adsorption is expected for the lower ratio, at least under dry conditions. Why having pretreated the samples in presence of water ? This makes the discussion complex because some acid sites may adsorb water before propene adsorption.

Conclusions

In view of the above remarks, the conclusions sound at least a bit too optimistic or maybe misleading.

Hence, the conclusions should be rewritten more carefully.

Reviewer #3 (Remarks to the Author):

Gao et al. reported the successful preparation of ZSM-5 nanorod arrays with a unique distribution of oriented pores and single crystalline array structures. In particular, the ZSM-5 nanorod arrays exhibit excellent performance in propene capture, meanwhile, the propene removal capacity could be obtained over the ZSM-5 array integrated with the Co₃O₄ nanoarray. However, I cannot recommend this work publication in Nature Communications in view of the innovation, and credibility of the conclusion. The reasons are as follows:

1. In the introduction section, the author introduced a lot of work “Specifically, in ZSM-5 (MFI framework, a classic type of zeolites with sinusoidal channels along a-axis and straight channels along b-axis)....”. However, the author falls short in elucidating the significance of this work and the pressing need for its exploration. It is crucial to provide a clearer explanation of why this research is important and the urgency underlying its pursuit.
2. The authors should elucidate the novelty of this research in comparison to previously reported works (10.1002/adfm.200305118; 10.1002/adfm.20040004).
3. Why not load metal onto the prepared ZSM-5 film and subsequently initiate the propane dehydrogenation reaction, given its excellent adsorption capacity for propane?
4. Additional experimental details regarding propane oxidation should be provided, including whether the catalyst requires pretreatment and the specific conditions of such pretreatment

Reviewer #4 (Remarks to the Author):

This work presents a unique and interesting design on oriented arrays of single crystalline microporous ZSM-5 in monolithic form for enhancing the small molecule capture dynamics and capacity. The C-oriented nanorod arrays of ZSM-5 directly grown on the ceramic honeycomb monolith were demonstrated with an advantage for faster transport of oligomerization reactants in and products out of the porous zeolites. An obvious mitigation of long-chain HCs and coking formation was accomplished in the multifunctional HC sorption/oxidation devices. This could be a generic merit for such array structured microporous film design as solutions to overcome these materials' constriction in sorption and reaction kinetics of small molecules in many anisotropic solid porous materials used in chemical and petrochemical production industries. The results and discussions of this work have been well presented with sound details. The work could be accepted after addressing the comments below.

1. The C-oriented array design is well versed, with sound proof and clearly-advantageous demo in the preferred fast transient sorption and reaction kinetics to benefit the adsorption, desorption, and

conversion functions that may be involved in the multifunctional devices. How about other orientations? It would be helpful to discuss further with respect to the choices of guest molecules and host absorber structure design at the end.

2. Fig. 2 as of now is a bit unclear in the labelling, especially in (b) and (c), where the use of 57.6% and ratio labels is a bit confusing, similarly the units in the Table 1 are unclear due to the tabulate format issue. The authors should consider revise and improve the clarity of these data presentations throughout.

3. Normally 'symbolic t' is used for the time parameter nomenclature wise, but 't' is directly used in the manuscript, the authors may consider adopt 'symbol t' instead.

4. There are quite some wording and grammar issues the authors should take care in order to make the flow better. For instance, the 'HZ-20-A' is used in singular tense mostly in the manuscript, but in some occasions plural tense is used. For example, in the discussion paragraphs over Fig. 2, it was spotted '!.. HZ-20-A yield less CO₂ and lower percentage of total generated CO₂ above 550 °C HZ-80-F sample'. In another occasion, 'The enrichment of captured C₃H₆..., which was also benefit to the adsorption process. However, ...if they could not be timely diffused out due to a potential diffusion barrier from zeolite micropores and block the accessibility to the acid sites.' The authors could make corrections here, say 'beneficial' in place of 'benefit', and '!...could not timely diffuse out..!' in place of '!...could not be timely diffused out..!'

5. Some typos and errors in the manuscript should be corrected, such as the reference error messages seemly occurred during the formatting.

6. What is the next step given the novel design in the array structured absorber films? Some in-depth discussions over the structure arrangement, spacing, dimensions, and even the compositions, would be really worthwhile to help point to broadly engaged directions to the readers.

Point-to-point Responses to Reviewers' Comments

Responses to Reviewer #1:

The manuscript discusses an innovative approach to enhance the adsorption and desorption properties of zeolites for cold trap applications. The process of adsorbing hydrocarbons at low temperatures and releasing them at higher temperatures is crucial for vehicle emission control, especially considering the low activity of catalysts during cold start periods.

The concept presented in the manuscript is both intriguing and novel. Through the manipulation of the orientation of zeolite (MFI) crystals, the authors achieved an enhanced adsorption rate. This adjustment in adsorption kinetics played a vital role in minimizing oligomer formation, showcasing the effectiveness of the proposed method.

Notably, the authors conducted tests under conditions closely resembling real-world applications, a rarity in research focused on the development of cold trap adsorbents. Consequently, I recommend the publication of this work, contingent upon the authors addressing the following technical concerns:

Response: We thank the reviewer for the strong endorsement of this work. Below are our point-to-point responses to the technical concerns.

1. The manuscript lacks clarity on how the authors controlled the orientation of zeolite crystals. Although the synthesis methods indicate variations in Si/Al ratios and NaOH concentrations in the two zeolite samples, the authors should provide insights into the control of orientation.

Response: We thank the reviewer for the comment. The secondary growth with the assistance of the pre-deposited seeds has been demonstrated as a robust approach for MFI membrane orientation control, as it decouples the nucleation stage from the crystal growth process as the following stage. The growth of zeolite films on the substrates follows a competitive growth model, namely "evolutionary selection"^{1, 2, 3}. The pre-deposited seeds in a random orientation firstly grow larger to form an initial continuous film, then the crystals intergrow with each other adjacently, while the crystal growth will be dominated with the orientations of the fastest growth rate. For MFI crystals, *c*-orientation usually is the fastest growth direction, followed by *b*-orientation and then *a*-orientation. As a result, ZSM-5 films usually end up with a *c*-oriented columnar structure given sufficient crystallization time. This theory has been stated in the Supplementary Materials to explain the different growth habits for HZ-20-A nanoarray and HZ-80-F film. It is noted that other factors may also impact the film orientation, such as the seeding step factors⁴ including the stacking behavior, density and orientation of pre-coated seeds, and the factors in the secondary-growth stage⁵ including the temperature, growth time and chemical composition of the synthesis solution. In this work, on the 3D monolithic cordierite surfaces, the seeds were deposited in a random orientation distribution, while the following ZSM-5 film crystal growth is dominated by *c*-orientation given the sufficient duration of crystal growth after 24h at 180 °C.

2. Given the significant roles of Si/Al ratios in oligomer formation and desorption temperature, I suggest exploring the creation of ZSM-5 with Si/Al=80 and a *c*-oriented array. This would enable an independent study of the effects stemming from both Si/Al ratio and orientation.

Response: We thank the reviewer for the insightful comment. Accordingly, we have been looking into the influences of various parameters from seeding step to secondary growth step, and exploring the orientation and structure control of ZSM films with varying Si/Al ratios, including the 80 one utilized in this work.

Based on our studies so far, we have identified that alkalinity as a critical parameter to synthesize and tailor the growth habit of ZSM-5 nanoarray. As shown in Figure R1, samples have been prepared with $\text{NaOH}/\text{SiO}_2 = 0, 0.2, 0.4, 0.6$ and 0.8 at $\text{Si}/\text{Al}=20$, and $\text{NaOH}/\text{SiO}_2 = 0, 0.2$ and 0.6 at $\text{Si}/\text{Al}=80$. Figure R1(d) and (f) show the SEM image of ZSM-5 nanoarray and film reported in the manuscript, respectively. For $\text{Si}/\text{Al}=20$, tuning of the crystal size and well-spaced nanoarray can be achieved through NaOH/SiO_2 ratio from 0 to 0.6. Upon further increase of NaOH/SiO_2 from 0.6 to 0.8, large zeolite particles are formed on the cordierite substrate surface with a significant occurrence of homogenous crystallization under such a high alkalinity. $\text{NaOH}/\text{SiO}_2 = 0.6$ is an optimum ratio for nanoarray structure formation at $\text{Si}/\text{Al}=20$. On the other hand, for $\text{Si}/\text{Al}=80$, the prepared samples present crystals intergrowth and tend to form a continuous film morphology with 0 and 0.2 NaOH/SiO_2 ratio. While, at $\text{NaOH}/\text{SiO}_2=0.6$, the anisotropic crystal growth seems to be retarded by the high alkalinity, leading to the formation of array structure of isotopically grown ZSM-5 crystal pyramids (Figure R1(h)). Nevertheless, at the higher Si/Al ratio scenario, the intergrowth phenomena seem to be much more dominant across the pH range, which may indicate the need of fine adjustment of other thermodynamic and kinetic parameters such as temperature and selective surface passivation to boost anisotropic growth of the ZSM crystals individually toward high-aspect ratio array structure formation.

In summary, the exploration and understanding of the impact of different synthetic factors is ongoing on the synthesis and control of oriented array structured ZSM-5 film. Specifically, we continue to investigate systematically the thermodynamic and kinetic parameters on the seeding and intergrowth of ZSM-5 nanoarray structure at various Si/Al ratios. With more fine adjustment of the growth kinetics and thermodynamics, especially through other parameters such as temperature and surface passivation, the array structured film of different orientations could be obtained for our future studies.

Figure R1. SEM images of ZSM-5 samples synthesized with different secondary-growth solution after 24h at 180 °C. (a-e) Si/Al=20 with NaOH/SiO₂=0/0.2/0.4/0.6/0.8, and (f-h) Si/Al=80 with NaOH/SiO₂=0/0.2/0.6.

3. Considering the desorption temperature exceeding 500°C for complete oligomer desorption, there is a concern about its practicality for cold trap applications. The authors should address whether this temperature is too high for such applications.

Response: We thank the reviewer for the comment. It is noted, the current propene adsorption and desorption performance in Figure 2a were evaluated on proton-exchanged ZSM-5 samples without any active metal loading such as Pt, Pd, Cu, etc. As the focus of current work is to demonstrate the advantages of our unique ZSM-5 nanoarray structure in adsorption and reaction dynamics, we intentionally focus only on the exchanged ZSM-5 samples with proton and separate the interaction of other metals from the current study. The desorption temperature and oxidation temperature will be significantly dropped once the active metals are loaded, which has been clearly revealed in the propene oxidation performance shown in Figure 3 when combining the proton-exchanged ZSM-5 nanoarray with Co₃O₄ nanoarray as oxidation catalysts.

On the other hand, the desorption temperature of 500 °C for the propylene oligomerization products is relatively high for car exhaust temperature and VOCs abatement. As shown in the experimental results Figure 2a, a large portion of CO₂ is generated at temperature lower than 500 °C. Given the fact that oligomerization of propylene on zeolite can happen even at room temperature⁶, the carbon length of propylene oligomerization products can vary significantly and is highly affected by the zeolite crystallite size, Si/Al ratio, guest ions and pore structures. It is known that hydrocarbons of longer carbon chain tend to have a higher boiling point. Therefore, effective methods to mitigate oligomerization reaction and oligomerization extent are possible routes to decrease the product desorption temperature and meet the challenge.

Many efforts have been conducted for mitigating the oligomerization reactions. Elizabeth E. Bickel et al.⁷ reported that propene dimerization rates decrease monotonically with increasing crystallite size for MFI zeolites synthesized with fixed H⁺-site. While, M. Bernauer and co-workers found that the proximity of protons (~5 Å) derived from the distribution of Al atoms in the H-ZSM-5 framework can substantially affect the rate of propene oligomerization⁸. Besides, the H-ZSM-5 zeolites with narrow intersecting pores can also provide a shape-selective effect advantageously limiting the formation of bulky aromatics and polyaromatic coke⁹.

Thus, based on the current work and understanding, further modification of the ZSM-5 crystallite size, intercrystallite space and Si/Al ratio could be future directions to work on to tackle the high desorption temperature.

References:

1. Weng J, Zhao B, Suib SL, Gao P-X. Oriented MFI films for gas phase separation, catalysis, and sensing: A review of crystal growth, design, and function enabling. *MRS Communications* **13**, 725-739 (2023).
2. Bonilla G, Vlachos DG, Tsapatsis M. Simulations and experiments on the growth and microstructure of zeolite MFI films and membranes made by secondary growth. *Microporous and Mesoporous Materials* **42**, 191-203 (2001).
3. Fu D, van der Heijden O, Stanciakova K, Schmidt JE, Weckhuysen BM. Disentangling Reaction Processes of Zeolites within Single-Oriented Channels. *Angewandte Chemie International Edition* **59**, 15502-15506 (2020).
4. Yan L, et al. Improved para-Xylene Selectivity in meta-Xylene Isomerization Over ZSM-5 Crystals with Relatively Long b-Axis Length. *ChemCatChem* **5**, 1517-1523 (2013).
5. Lai Z, Tsapatsis M, Nicolich JP. Siliceous ZSM-5 Membranes by Secondary Growth of b-Oriented Seed Layers. *Advanced Functional Materials* **14**, 716-729 (2004).
6. Hawkins AP, et al. Low-temperature studies of propene oligomerization in ZSM-5 by inelastic neutron scattering spectroscopy. *RSC Advances* **9**, 18785-18790 (2019).
7. Bickel EE, Gounder R. Hydrocarbon Products Occluded within Zeolite Micropores Impose Transport Barriers that Regulate Brønsted Acid-Catalyzed Propene Oligomerization. *JACS Au* **2**, 2585-2595 (2022).
8. Bernauer M, et al. Proton proximity – New key parameter controlling adsorption, desorption and activity in propene oligomerization over H-ZSM-5 zeolites. *Journal of Catalysis* **344**, 157-172 (2016).
9. Müller S, et al. Coke formation and deactivation pathways on H-ZSM-5 in the conversion of methanol to olefins. *Journal of Catalysis* **325**, 48-59 (2015).

Responses to Reviewer #2:

In this paper, the author wished to demonstrate the superior adsorptive and catalytic performances of a ZSM-5 film specifically designed with nanocrystals oriented perpendicular to the substrate as compared with a more conventional film with a crystal orientation parallel to the surface. It is proposed that the peculiar array structuration of the nanocrystals both enhances the sorption kinetics and sorption capacity of propene while decreasing coking effects during desorption. The studied context is that of cold-start application where such requirements are important.

Although this research is important for the adsorption and catalysis community and was rather well carried out in many aspects, some issues remained to be addressed before to consider the manuscript for publication. Overall, I am not fully convinced of the presented performances and related explanations even if it clear that nanostructuration may be helpful in achieving better performances.

It seems difficult to compare the performances from samples having a different Si/Al ratio because the latter parameter is very influent on propene adsorption, especially. Other problems are related to the testing methodologies (breakthrough tests and DRIFTS). The DRIFTS section is not useful because the data seem not very reliable and not interpreted correctly.

Response: We thank the reviewer for recognizing the importance of this work, as well as the constructive comments and concerns detailed below. We have provided our point-to-point responses to the comments.

Specific comments are listed below:

General:

- All along the manuscript, there are many problems related to the insertion of links to the figures in the pdf file (error - reference source not found).

Response: We thank the reviewer for the comment. Accordingly, we have corrected the errors of links to the figures and references source in the revision.

- Is there any specific reason for the choice of ZSM-5 as preferred zeolite for the trapping of propene? Although this zeolite was demonstrated to be interesting for cold-start application, its porosity is made from interconnecting straight and zigzag channels. Hence, it could possibly be more complex to address the effect of pore orientation as compared with pseudo-1D zeolites such as mordenite, which is also important for cold-start as shown for instance in refs. 28 and 29.

Response: We thank the reviewer for the comment. MFI, MOR, BEA, FAU, and FER zeolites are known as the “Big Five” zeolites¹. They are all extensively used in the present chemical and petrochemical industries, especially for petrochemical production. The reason of choosing ZSM-5 in this study is that ZSM-5 zeolite with two channels that intersect each other and form a three-dimensional channel system can not only be used as an effective adsorbent, but also as a catalyst platform². It is widely employed in important processes and applications across of the chemical and petrochemical industries, such as hydrocarbon chemical and fuel manufacturing including methanol to gasoline and diesel (MTX) process, and oil refining.

In addition, as reported in refs. 28 and 29, H-MFI showed an excellent trapping efficiency of HCs, especially propene, as captured in both zigzag (*a*-) and straight (*b*-) channels. In this work, we intended to maximize the exposure of *a*- and *b*- channels on a *c*-oriented ZSM-5 nanoarray structure to facilitate the mass transportation and trapping site utilization. We agree that mordenite with the pseudo-1D micropore structure could be another promising model to work with, considering its simple pore structure and wide applications. This material system is being included in our future study to help generalize current method, expand our learnings and formulate a suite

of knowledge base and tools on the synthesis, characterization, and application for different types of nanostructured zeolite.

Catalyst preparation and characterization:

- Although interesting, the effect of the Si/Al ratio on the film structuration is not very clear. At the bottom of page 4, the authors suggest that the higher concentration of Al during synthesis of Z-20-A (as compared with Z-80-F) is responsible of the differences observed, presumably through a repulsion mechanism in the initial stages of crystal growth. In that respect, it would be important to examine other Si/Al ratio to confirm this. Alternatively, it is also suggested in the general literature that anisotropic growth could be due to some adsorption of some specific species of the sides of rods, preventing their growth on corresponding directions.

Response: We thank the reviewer for the comment. We have revised the discussion to include both repulsion mechanism and surface passivation in the revised manuscript. Below are some elaborations accordingly.

The repulsion mechanism was reported by J. Caro et al.³ to explain the enhanced inter-crystalline defect transport on ZSM-5 membranes with increasing Al content. During the solution synthesis, the transport and attachment of the negatively charged MFI precursors to the similarly charged crystal surfaces result in the poor intergrowth of ZSM-5 crystals. By adding a so-called "Intergrowth Supporting Substance (ISS)" in the synthesis, the crystal surface is re-charged and improves the crystal intergrowth. Borrowing the similar idea but in the other direction, we demonstrated that the growth of ZSM-5 nanoarray was achievable and thus the adsorption dynamics were improved.

On the other hand, as surface passivation due to growth modifier addition can be utilized to induce anisotropic crystal growth with preferential orientation, resulting in the array structured film. R. Feng et al.⁴ prepared a *b*-axis oriented hierarchical ZSM-5 zeolite with glucose as the surface modifier on the (010) surface, resulting in a reduced growth rate along the *b* axis, and ZSM-5 with a *b*-axis of 220 nm can be synthesized. Similarly, Yu et al.⁵ discovered that in a microwave-assisted solvothermal synthesis system, -OH groups of diols could help decrease the surface energy of (010) and promoted the growth rate along *c*-axis. Besides, fluorine ion (F⁻¹) has been studied extensively for a short *b*-axis ZSM-5 synthesis, mainly due to the mineralization process with tunable solubility of silicon species^{6, 7}. For all these cited studies, thin crystals of thickness within nm range along *b*-axis were synthesized due to the capping effect, where the *b*-axis growth is significantly suppressed by surface passivation. While, in our work, the synthesized ZSM-5 nanoarray shows a thick and uniform *b*-axis growth with no sign of growth suppression. Instead, a well-spaced crystal allocation can be achieved along the *b*-axis, ensuring the maximum exposure of *a* and *b* channels.

In summary, in this work we demonstrated the successful synthesis of ZSM-5 nanoarray on the monolithic substrates and its advantages in adsorption dynamics, mass transport and coking mitigation. We have continued our investigation on the roles of various parameters during seeding (i.e., seeding approach, seeds size, seeds orientation, etc.) and secondary-growth stage (i.e., Si/Al ratio, alkalinity, surface passivation, solvent, etc.), fine tune and manipulate the growth control in terms of the crystal orientation, crystal size, nanoarray density distribution. Furthermore, we desire to expand the idea to other zeolites and explore a broader family of zeolitic nanostructures.

Please comment on these points.

- The N₂ adsorption isotherm of cordierite substrate has to be given and its porous characteristics have to be included in Table 1.

Response: We thank the reviewer for the comment. We have included the N₂ isotherm of cordierite substrates in Supplementary Figure 6 and summarize its porous characteristics in Table 1.

- Where are presumably located the mesopores in both films? They are not visible in Fig. 1. Could the improvement suggested by the authors on adsorption kinetics could be related to the presence of such mesopores rather than a different orientation of straight and zigzag channels?

Response: We thank the reviewer for the comment. The mesopores for the nanoarray structured ZSM-5 and film structured ZSM-5 are formed between the ZSM-5 single crystals (no mesopores are observed within ZSM-5 crystals themselves). As revealed in Figure 1, the inter-crystal spacing is much larger in the ZSM-5 nanoarray structure. Plus, the mesopore volume ($V_{\text{meso, sample}}$ in Table 1) is also much larger in the ZSM-5 nanoarray than the traditional continuous ZSM-5 film. This work aims to maximize the exposure of straight (*b*-axis) and zigzag (*a*-axis) channels for a faster adsorption pathway by expanding the inter-crystal spaces. Both the extra mesopores and *c*-orientation contribute to the improvement in adsorption and reaction dynamics of ZSM-5 nanoarray. Accordingly, we have elaborated the contribution of mesopores to the improvement of adsorption kinetics in ZSM-5 nanoarray in the revised manuscript, in the section "Catalysts Preparation and Characterization".

Performance Benefits Demonstration: Improved Reaction Kinetics – The amount and type of acid sites on both zeolite films, as measured from ammonia adsorption, is not commented in the text.

Response: We thank the reviewer for the comment. We have added the following discussions about the amount and type of acid sites on both zeolite films at the end of the section "Catalysts Preparation and Characterization".

"The acid sites of HZ-80-F and HZ-20-A were examined by NH₃-TPD as shown in Supplementary **Error! Reference source not found.** c-d. Both samples exhibited two major peaks in the temperature regions of 150-225 °C and 300-400 °C, which could be ascribed to the chemisorption of NH₃ species on weak and strong acid sites, respectively. With the Si/Al ratio decreasing from 80 to 20, the amount of both weak and strong acid sites increased remarkably. Accordingly, HZ-20-A possessed a much higher C₃H₆ adsorption capacity than HZ-80-F, because of its stronger adsorption affinity of unsaturated hydrocarbon due to a higher acidity. Meanwhile, the peaks of weak and strong acid sites shifted to a higher temperature from 184 °C and 326 °C in HZ-80-F to 207 °C and 360 °C in HZ-20-A, respectively."

However, it is about more than 10 times higher for Z-20-A as compared with Z-80-F, while the acid strength is not very different. Considering that acid sites are thought to be responsible for propene adsorption (as also shown in provided reference 28), it may be expected that the propene adsorption capacity may change by the same order of magnitude. However, it is not the case (Table 1). How do the authors explain this?

Response: We thank the reviewer for the comment. In the work, NH₃ temperature programmed desorption was used to quantify the acid sites of the prepared ZSM-5. Considering the kinetic diameter of NH₃ (260 pm) is much smaller than C₃H₆ (450 pm), it is reasonable that some of the NH₃ accessible acid sites are not available for C₃H₆ adsorption. Besides, C₃H₆ adsorption capacity can be affected by acid sites, while acid site is not the only factor. C₃H₆ adsorption is also highly related to the pore volume^{8,9}. As shown in Table 1, the nanoarray and film ZSM-5 have mesopores around 0.028 and 0.019 cm³g⁻¹ respectively, showing a relatively consistent trend with the C₃H₆ adsorption, indicating that the pore size here plays an important role for zeolite-C₃H₆ interaction.

- I find that the shape of breakthrough curves in Fig. 2 and Fig. S7 is a bit odd, as a period of total retention of propene is never reached for Z-80-F. Hence, I'm not sure that the methodology used to record the data is accurate enough.

Response: We thank the reviewer for the comment. First of all, a blank cordierite was also tested in the same reactor following the same procedure as a reference point against the other two ZSM-5 samples. As shown in Supplementary Figure 7 as the blue curve, blank cordierite could barely capture propene due to its low porosity (as revealed in Supplementary Figure 6), leading to a very fast recovery of propene signal after the feed gas was switched into the reactor from bypass. The recovered propene signal matched the original bypass level 333ppm and maintained over the rest time of the breakthrough test, which validated the reliability of our methodology for propene adsorption breakthrough test. Secondly, as mentioned in the comment and Supplementary Figure 7, both HZ-80-F and HZ-20-A could not reach the original propene level. One potential reason is that the breakthrough test was only conducted for 120 min. We observed the superior performance of propene adsorption on HZ-20-A over HZ-80-F, and believed the breakthrough testing over 120 min was sufficient to demonstrate it. At the end of 120-min testing, we could find the detected C_3H_6 signal still approach the original level of 333ppm but in a slower rate. It is reasonable to anticipate that the propene will eventually saturate the samples if we extend the test for a longer duration.

To conclude, we had verified our methodology of breakthrough testing via a reference testing on blank cordierite. We completed the breakthrough testing at 120 min with sufficient evidence collected to showcase the advantages of ZSM-5 nanoarray in adsorption kinetics. It is unnecessary to extend the breakthrough testing to a longer duration to observe the full recovery of propene adsorption on the two ZSM-5 samples.

For dynamic adsorption tests, how are the samples introduced in the tube to ensure that the whole flow is passing through and not on the sides?

Response: We thank the reviewer for the comment. The reactor setup for performance evaluation (propene adsorption test, cold-start test, ammonia temperature-programmed desorption test, etc.) is well-established in our laboratory and examined in our past research and publications^{10, 11, 12}. In general, "each sample was wrapped with a catalyst cushion mat and then loaded into the center of quartz tube (1 in. diameter). The cushion mat will be thermally expanded after high-temperature exposure to fill up the gaps inside the quartz tube and force the whole gas flow to only pass through the monolith". We have updated the above information in the manuscript.

What is the sampling interval to measure propene concentration all along the breakthrough?

Response: We thank the reviewer for the comment. The sampling interval to measure propene concentration all along the breakthrough testing is ~ 26 s. We have updated this information in the Supplementary Materials "Section S2: Characterization and Performance Test".

Why using a temperature of 100°C instead of a lower temperature to ensure that adsorption processes are not occurring together with catalytic ones.

Response: We thank the reviewer for the comment. As stated in the manuscript, the prepared ZSM-5 samples were evaluated as candidates for diesel exhaust cold start application. Therefore, Low Temperature Storage Catalyst Test Protocol created by USDRIVE¹³ (as shown in Figure R2) with a modified gas composition was adopted. Storage characterization isothermally at a default temperature of 100 °C for 30 min right after an initial pre-treatment at 600 °C for 20 min with 6 % H_2O were required by the Protocol as a standard procedure.

[REDACTED]

Figure R2. Test strategy and temperature control for storage catalyst test protocol by USDRIVE¹³

- The points could be shown on breakthrough curves and more details on the experimental methodology have to be added in the supplementary section. A scheme of the adsorption setup has to be given likewise some details about FTIR (number of scans, sampling interval, cell volume etc....).

Response: We thank the reviewer for the comment. We have added more details in the revision about FTIR instrument and parameters for gas measurement. The following information is added in the Supplementary Materials "Section S2: Characterization and Performance Test".

"After being calibrated for C₃H₆, CO, and CO₂, FTIR (Nicolet 6700, Thermo Scientific; equipped with 2 Meter Gas Cell in 200 ml internal cell volume) was used to detect the concentration of inlet and outlet gases. FTIR was set with number of scans of 16, resolution of 0.5, data spacing of 0.241 cm⁻¹, and estimated sampling interval of 26 s."

In addition, a scheme of the reactor for propene adsorption and cold-start test (as shown in Figure R3) is added to the revised manuscript.

Figure R3. Schematic of the reactor for propene adsorption and cold-start test.

- Considering these last remarks, I am not very convinced about the discussion on adsorption kinetics given on the top of page 7 likewise that on desorption behaviour on the same page. In my opinion, the desorption/oxidation profiles are very similar for both zeolites.

Response: We thank the reviewer for the comment. So far, we have provided comprehensively point-to-point responses to the above-cited reviewer's comments, regarding the explanation of characterization results (i.e., acidity from NH₃-TPD), sampling method, instrument setup (i.e., FTIR parameters), and standard protocols for performance evaluation (i.e., propene adsorption and cold-start). In our opinions, the superior advantages of ZSM-5 nanoarray structure over ZSM-5 film are well displayed based on their distinct behaviors in propene adsorption and desorption as shown in Figure 2. Specifically, in the 30 min adsorption stage (Figure 2 a-c), when the sample was exposed to propene, HZ-20-A completely removed propene from the feeding gas in a much faster rate than HZ-80-F, demonstrating the benefits of well-spaced nanoarray structure and maximized exposure of *a*- and *b*- channels in promoting the adsorption kinetics. The abundant acid sites on HZ-20-A also increased the propene capture capacity by ~57.6% as compared to HZ-80-F.

Furthermore, the different distribution of the species detected during desorption step (Figure 2d) was also evident to highlight the ZSM-5 array structure. Considering the acid sites of ZSM-5 plays a role in the oligomerization of captured propene, HZ-20-A with a much larger amount of acid sites was supposed to have a more intensified propene oligomerization and form more long-chain hydrocarbons and even coke. Surprisingly, HZ-20-A released less CO₂ and a lower percentage of total generated CO₂ above 550 °C (Figure 2d) although it contains more oligomerized products as revealed from TGA analysis (Supplementary Figure 8). This contradiction in between is attributed to the unique array-structured morphology of HZ-20-A that helps mitigate the long-chain HCs and coking formation.

Finally, we also proved the mechanical robustness of the synthesized HZ-20-A (Supplementary Figure 12), and reproducibility of the synthesis approach and performance evaluation results (Supplementary Figure 13), which are other critical factors in a practically relevant application.

In summary, we have comprehensively characterized and demonstrated the advantages of unique ZSM-5 nanoarray structures in promoting the adsorption and reaction dynamics from a logical, reliable and repeatable perspective.

- DRIFTS page 10. The assignment of infrared bands has to be done with much more accuracy, starting from propene in the gas phase and then coming back to adsorbed species.

What are the bands located at 1300 and 1175 cm⁻¹?

Response: We thank the reviewer for the comment. Accordingly, we have added the following description in the revised manuscript to include the fine details of the FTIR spectrum analysis across different bands.

“At 100 °C, a set of bands can be observed that belong to propene itself and propene generated oligomers. In the low-IR range from 1100 to 1500 cm⁻¹, peak at 1200 cm⁻¹ is from the =CH₂ stretching of gas phase C₃H₆¹⁴. Peaks ranging from 1320-1360 cm⁻¹ are due to aromatic ring C-C stretching. While 1500 cm⁻¹ is the C=C from the aromatic like compounds¹⁵, the peak at 1470 cm⁻¹ is caused by the vibration of H-C₃H₆ adsorbed via their π bonds¹⁶. In addition, at high-IR range from 2700 to 3100 cm⁻¹, -CH₃ symmetric stretching (2933 cm⁻¹), asymmetric stretching (2960 cm⁻¹), and -CH₂- stretching (2868 cm⁻¹) modes of oligomeric species can be observed¹⁵.”

The detailed IR peaks related to propene adsorption are summarized in the Table R1.

Table R1. Wavenumbers of adsorbed species upon propene adsorption over H-ZSM-5 Zeolite¹⁵

wavenumber/cm ⁻¹	adsorbed species	functional group	vibrational mode
1415	propene (π-complex with BAS)	=CH ₂	δ(CH ₂)
1455		-CH ₃	δ _a (CH ₃)
1633		-C(H)=CH ₂	ν(C=C)
1369	propene oligomers	-CH(CH ₃) ₂	δ _s (CH ₃)
		-C(CH ₃) ₃	
1382		-CH ₃	
		-CH(CH ₃) ₂	
		-C(CH ₃) ₂ -	
1393		-C(CH ₃) ₃	
1467, 1469		-CH ₃	δ _a (CH ₃)
		-CH ₂ -	δ(CH ₂)
1620–1660		-C=C-	ν(C=C)
2860	propene and propene oligomers	-CH ₃ , -CH ₂ -	2δ _a (CH ₃), ν _s (CH ₃), ν _s (CH ₂)
2930–2960			ν _s (CH ₃), ν _a (CH ₃), ν _a (CH ₂)
1510	alkyl-substituted CPCs	[>C=C(H/CH ₃)- C<] ⁺	ν _{as} (C=C-C)
1495	benzene-like compounds	aromatic ring	ν(C=C)
1610			
1320–1360	condensed aromatic species	aromatic ring	ν(C-C)
1540			ν(C=C)

What about the C=C stretch located around 1650 cm⁻¹ and the evolution of acid O-H groups? Note that the 1470 cm⁻¹ is one of C-H bending modes.

Response: We thank the reviewer for the comment. The extended IR spectrum at 1100-1700 cm⁻¹ is shown in Figure R4. As can be seen in Figure R4, the C=C stretch peak around 1650 cm⁻¹ is hardly to be observed, which is due to the low C₃H₆ partial pressure used (only 333 ppm C₃H₆

was fed into DRIFTS chamber)¹⁷. Instead, C-C and C-H signals from the oligomerization products can be clearly viewed after 30 min propene feeding, indicating a strong oligomerization pathway. This is also reflected from the acid O-H groups evolution in the range of 3550-3900 cm^{-1} shown in Figure R5. As can be observed, the O-H groups of the ZSM-5 decrease along with the propene adsorption and oligomerization which is due to the increased coverage by the oligomerization products.

Figure R4. In-situ diffuse reflectance infrared Fourier transform spectroscopy (DRIFTS) spectra of proton-exchanged (a) conventional ZSM-5 film (HZ-80-F) and (b) array-structured ZSM-5 film (HZ-20-A) during the propene adsorption under 333 ppm $\text{C}_3\text{H}_6 + 12\% \text{O}_2 / \text{N}_2$ at 100 °C for 30 min at the wavenumber range 1100-1700 cm^{-1} .

Figure R5. In-situ diffuse reflectance infrared Fourier transform spectroscopy (DRIFTS) spectra of proton-exchanged (a) conventional ZSM-5 film (HZ-80-F) and (b) array-structured ZSM-5 film (HZ-20-A) during the propene adsorption under 333 ppm $\text{C}_3\text{H}_6 + 12\% \text{O}_2 / \text{N}_2$ at 100 °C for 30 min at the wavenumber range 3550-3900 cm^{-1} .

- Moreover, the profiles of DRIFT spectra are influenced by the zeolite absorption, which is very strong below 1600 cm^{-1} . From the reviewer experience, the bands of adsorbed species could be very much perturbed and it is not possible to do quantitative measurements by DRIFTS. Transmission with pellets if of greater help here. For comparison and better interpretation, the authors should pay attention to J. Phys. Chem. C 2018, 122, 11, 6128–6136.

Response: We thank the reviewer for the comment. We considered running transmission with pellets here at the beginning. However, given a second thought that the most highlight of our work was to design and prepare unique ZSM-5 nanoarray structure with well-spaced individual crystals and maximized exposure of ZSM-5 *a*- and *b*- channels, the nanoarray structure would be destroyed during the sample preparation for pellets experiment. Thus, we turned to run the DRIFTS on the monolithic samples with the intact nanoarray structure maintained.

- I don't understand the trends observed for the washcoated samples in function of Si/Al ratio. Higher propene adsorption is expected for the lower ratio, at least under dry conditions.

Response: We thank the reviewer for the comment.

The propene adsorption capacity of ZSM-5 is not only determined by the amount of acid sites, but also highly influenced by the pore volume and pore size^{8, 9}. The texture characteristics of washcoated HZP-87 and HZP-26 are added in the manuscript supplement. As shown in Supplementary Table 2, the washcoated HZP-87 has a higher external surface area while the total pore volume is similar, leading to a potential more accessible adsorption site for propene guest molecules than HZP-26. In addition, as claimed in the manuscript, propene oligomerization could occur easily on the ZSM-5 acid sites at 100 °C. The produced oligomerization products may block the adsorption sites for propene if it could not be diffused out timely, which is also affected by the texture properties of washcoated samples. Therefore, although the amount of acid sites of HZP-26 is larger than that of HZP-87, HZP-26 is still reasonable to exhibit a lower propene adsorption capacity over HZP-87.

Why having pretreated the samples in presence of water? This makes the discussion complex because some acid sites may adsorb water before propene adsorption.

Response: We thank the reviewer for the comment. As stated in the manuscript, the prepared ZSM-5 samples were evaluated as candidates for diesel exhaust cold start application. Therefore, Low Temperature Storage Catalyst Test Protocol created by USDRIVE¹³ (Figure R2) with a modified gas composition was adopted. Storage characterization isothermally at a default temperature of 100 °C for 30 min right after an initial pre-treatment at 600 °C for 20 min with 6 % H₂O were adopted according to the Protocol.

Conclusions

In view of the above remarks, the conclusions sound at least a bit too optimistic or maybe misleading.

Hence, the conclusions should be rewritten more carefully.

Response: We thank the reviewer for the comment. We have revised the conclusions to make it more rigorously presented and aligned with the results and discussions.

References:

1. Narayanan S, Tamizhdurai P, Mangesh VL, Ragupathi C, Santhana krishnan P, Ramesh A. Recent advances in the synthesis and applications of mordenite zeolite – review. *RSC Advances* **11**, 250-267 (2021).
2. Lima CGS, Jorge EYC, Batinga LGS, Lima TdM, Paixão MW. ZSM-5 zeolite as a promising catalyst for the preparation and upgrading of lignocellulosic biomass-derived chemicals. *Current Opinion in Green and Sustainable Chemistry* **15**, 13-19 (2019).
3. Noack M, *et al.* Effect of crystal intergrowth supporting substances (ISS) on the permeation properties of MFI membranes with enhanced Al-content. *Microporous and Mesoporous Materials* **97**, 88-96 (2006).

4. Feng R, Yan X, Hu X, Zhang Y, Wu J, Yan Z. Phosphorus-modified b-axis oriented hierarchical ZSM-5 zeolites for enhancing catalytic performance in a methanol to propylene reaction. *Applied Catalysis A: General* **594**, 117464 (2020).
5. Chen X, Yan W, Cao X, Yu J, Xu R. Fabrication of silicalite-1 crystals with tunable aspect ratios by microwave-assisted solvothermal synthesis. *Microporous and Mesoporous Materials* **119**, 217-222 (2009).
6. Zhang J, *et al.* Tailored Synthesis of ZSM-5 Nanosheets with Controllable b-Axis Thickness and Aspect Ratio: Strategy and Growth Mechanism. *Chemistry of Materials* **34**, 3217-3226 (2022).
7. Qin Z, *et al.* Comparative Study of Nano-ZSM-5 Catalysts Synthesized in OH⁻ and F⁻ Media. *Advanced Functional Materials* **24**, 257-264 (2014).
8. Koyama T-r, *et al.* Key role of the pore volume of zeolite for selective production of propylene from olefins. *Physical Chemistry Chemical Physics* **12**, 2541-2554 (2010).
9. Zhu X, Liu S, Song Y, Xu L. Catalytic cracking of C4 alkenes to propene and ethene: Influences of zeolites pore structures and Si/Al₂ ratios. *Applied Catalysis A: General* **288**, 134-142 (2005).
10. Lu X, *et al.* Direct Synthesis of Conformal Layered Protonated Titanate Nanoarray Coatings on Various Substrate Surfaces Boosted by Low-Temperature Microwave-Assisted Hydrothermal Synthesis. *ACS Applied Materials & Interfaces* **10**, 35164-35174 (2018).
11. Lu X, *et al.* Ion-Exchange Loading Promoted Stability of Platinum Catalysts Supported on Layered Protonated Titanate-Derived Titania Nanoarrays. *ACS Applied Materials & Interfaces* **11**, 21515-21525 (2019).
12. Tang W, *et al.* Ceria-based nanoflake arrays integrated on 3D cordierite honeycombs for efficient low-temperature diesel oxidation catalyst. *Applied Catalysis B: Environmental* **245**, 623-634 (2019).
13. Rappé KG, *et al.* Aftertreatment Protocols for Catalyst Characterization and Performance Evaluation: Low-Temperature Oxidation, Storage, Three-Way, and NH₃-SCR Catalyst Test Protocols. *Emission Control Science and Technology* **5**, 183-214 (2019).
14. Busca G, Ramis G, Lorenzelli V, Janin A, Lavalley J-C. FT-i.r. study of molecular interactions of olefins with oxide surfaces. *Spectrochimica Acta Part A: Molecular Spectroscopy* **43**, 489-496 (1987).
15. Lashchinskaya ZN, Gabrienko AA, Prosvirin IP, Toktarev AV, Stepanov AG. Effect of Silver Cations on Propene Aromatization on H-ZSM-5 Zeolite Investigated by ¹³C MAS NMR and FTIR Spectroscopy. *ACS Catalysis* **13**, 10248-10260 (2023).
16. Bernauer M, *et al.* Proton proximity – New key parameter controlling adsorption, desorption and activity in propene oligomerization over H-ZSM-5 zeolites. *Journal of Catalysis* **344**, 157-172 (2016).
17. Rubeš M, Koudelková E, de Oliveira Ramos FS, Trachta M, Bludský O, Bulánek R. Experimental and Theoretical Study of Propene Adsorption on K-FER Zeolites: New Evidence of Bridged Complex Formation. *The Journal of Physical Chemistry C* **122**, 6128-6136 (2018).

Responses to Reviewer #3:

Gao et al. reported the successful preparation of ZSM-5 nanorod arrays with a unique distribution of oriented pores and single crystalline array structures. In particular, the ZSM-5 nanorod arrays exhibit excellent performance in propene capture, meanwhile, the propene removal capacity could be obtained over the ZSM-5 array integrated with the Co₃O₄ nanoarray. However, I cannot recommend this work publication in Nature Communications in view of the innovation, and credibility of the conclusion. The reasons are as follows:

Response: We thank the reviewer for the comments. We have provided our point-to-point responses to address the reviewer's main concern on the novelty of our work.

1. In the introduction section, the author introduced a lot of work "Specifically, in ZSM-5 (MFI framework, a classic type of zeolites with sinusoidal channels along a-axis and straight channels along b-axis)...". However, the author falls short in elucidating the significance of this work and the pressing need for its exploration. It is crucial to provide a clearer explanation of why this research is important and the urgency underlying its pursuit.

Response: We thank the reviewer for the comment. Accordingly, we have expanded our reasoning in the introduction on the significant role of manipulating ZSM-5 orientation to fit the different application requirements such as separation, catalytic activity, product selectivity and adsorption, which is the first key point we want to emphasize before we propose the unique array structure as a solution to address the kinetic constraints of adsorption and reaction. When it turns to adsorption application, although the extensive efforts have been made to improve the uptake level of guest species, much less attention was given to the kinetic restrictions and transient reactions within a short duration in a practical environment. The current work aims to fill the gap and points out a possible route to push the ZSM-5 catalytic applications one step further, combining with faster kinetics and high capacity, through well-designed and tailored ZSM-5 orientation and nanoarray structure.

2. The authors should elucidate the novelty of this research in comparison to previously reported works (10.1002/adfm.200305118; 10.1002/adfm.200400040).

Response: We thank the reviewer for the comment. The highlight of the work is that we successfully designed a unique distribution of oriented pores and well-spaced c-oriented ZSM-5 array structure on a honeycomb monolith. With the well-controlled spacing, ZSM-5 straight channels along b-axis and zig-zag channels around a-axis can be maximumly exposed, resulting in the increased propene adsorption kinetics and capacity. Besides, we also demonstrated the synthesis and mechanical robustness of the zeolite-monolith assembly as a whole piece.

For Geon-Joong Kim et al.'s work¹, powder-form ZSM-5 with a bimodal micro/mesoscopic pore system was prepared by using a solid rearrangement process using the framework of a tetrapropylammonium hydroxide (TPAOH) impregnated mesoporous precursor such as MCM-41 or SBA-15 as template with nanostructured carbons in the pore channels. The mesopores generated are mainly inside the zeolite crystals which is much different with the inter crystal spacing we are controlling in this work. Besides, despite more mesopores are generated, the powder form ZSM-5 crystals synthesized are still in a random order in the microscale, in distinct contrast with the well-defined nanoarray structure we demonstrated.

Meanwhile, for Jeffrey P. Nicolich's work², a b-oriented siliceous ZSM-5 membrane was prepared using a modified seeded growth procedure. A membrane configuration is formulated with b-axis oriented straight channels as the surface. In this way, the prepared film can have reduced gaps and space between different ZSM-5, resulting in fewer defective crystals, and become more robust to crack formation with decreased leakage and improved separation performance.

Nevertheless, the membrane formed is composed of many overlapping ZSM-5 crystals with possibly low accessibility of *a*-axis zig-zag channels. This is very different from the well-spaced nanoarray structure presented in our work that is focused on controlling the ZSM-5 crystals' spacing and maximizing both exposure of *b*-axis and *a*-axis channels for improved adsorption kinetics and capacity.

3. Why not load metal onto the prepared ZSM-5 film and subsequently initiate the propane dehydrogenation reaction, given its excellent adsorption capacity for propane?

Response: We thank the reviewer for the comment. In this work, we have demonstrated the feasibility of our adsorber design to overcome both sorption and reaction kinetic restrictions of the porous materials that possess anisotropic channels. By tailoring its crystal orientation and building novel nanoarray structure, ZSM-5 nanoarray structured film (HZ-20-A) exhibited superior advantages in propene adsorption and coking formation mitigation. Although the reviewer may have mistaken the propene with propane in this comment, it is indeed an interesting topic to explore our ZSM-5 nanoarray structure in the propane dehydrogenation (PDH). We have conducted experiments accordingly with some interesting results presented below, which clearly point to the superior sorption kinetic advantage showcased by the array structured ZSM-5.

1 wt.% of Pd loaded samples based on Z20A (array structure, denoted as Pd-Z20A) and washcoated ZSM-5 with similar Si/Al ratio (denoted as Pd-WC20) were prepared for PDH. The loading mass of Z20A and WC20 on monoliths were controlled at a similar level to ensure the catalyst mass were comparable. Pd-WC20 was used as a bench mark here to showcase the structure advantages of Pd-Z20A. Both catalysts were evaluated for PDH at 600 °C for 12h after a reduction pretreatment at 600 °C for 1 h with 5% H₂ in Ar. As shown in Figure R6, a two-range PDH performance can be observed. In the first 100 min, a transition period presents with a decrease in C₃H₈ conversion and an increase in C₃H₆ selectivity. Then C₃H₈ conversion and C₃H₆ selectivity reach a steady state and showcase good performance stability throughout the rest testing time. It is noted that C₃H₈ conversion and C₃H₆ selectivity are comparable (8% C₃H₈ conversion and 40% C₃H₆ selectivity) for Pd-Z20A and Pd-WC20 catalysts, indicating a similar PDH performance. The major difference between the two catalysts is that Pd-Z20A shows a shorter transition period and reaches steady state much faster compared to the Pd-WC20 counterpart. This is in highly agreement with the fast kinetics performance of propene adsorption experiments shown in the manuscript. The preliminary PDH experiment shown here clearly demonstrates the structure advantages of Pd-Z20A, pointing out that the prepared Pd-Z20A could be a potential promising catalyst for PDH. Despite this promising demonstration of PDH advantage in the Pd supported on array structured ZSM-5, to maintain the conciseness, thoroughness, and the focus of this work, we decide not to add these interesting data in the revised manuscript. Nevertheless, the Pd loaded ZSM-5 catalyst structure, composition, and reaction parameters will be further tuned and studied in detail to achieve the optimum PDH performance in our future work.

Figure R6. Summary of propane dehydrogenation (PDH) in terms of C_3H_8 conversion and C_3H_6 selectivity using Pd-Z20A and Pd-WC20 (washcoated ZSM-5 with Si/Al ratio of 20). For both catalysts, ~1 wt.% of Pd was loaded by using dip-coating method. The catalyst was firstly reduced at 600 °C for 1 h with 5% H_2 in Ar. PDH reaction was conducted at 600 °C for 12 h, using 5% of C_3H_8 in Ar, with a GHSV of $12,000\ h^{-1}$.

4. Additional experimental details regarding propane oxidation should be provided, including whether the catalyst requires pretreatment and the specific conditions of such pretreatment.

Response: We thank the reviewer for the comment. It is noted that propane oxidation performance was not evaluated in this work. Instead, we focused on the dynamic adsorption of propene in the manuscript, and performed the propene adsorption testing and its co-benefits in cold-start application combining with a well-reported oxidation catalyst Co_3O_4 nanoarray. The testing details of both propene adsorption and cold-start application, including the required treatment and specific conditions, were described in the Supplemental Materials.

References:

1. Cho SI, Choi SD, Kim J-H, Kim G-J. Synthesis of ZSM-5 Films and Monoliths with Bimodal Micro/Mesoscopic Structures. *Advanced Functional Materials* **14**, 49-54 (2004).
2. Lai Z, Tsapatsis M, Nicolich JP. Siliceous ZSM-5 Membranes by Secondary Growth of b-Oriented Seed Layers. *Advanced Functional Materials* **14**, 716-729 (2004).

Responses to Reviewer #4:

This work presents a unique and interesting design on oriented arrays of single crystalline microporous ZSM-5 in monolithic form for enhancing the small molecule capture dynamics and capacity. The C-oriented nanorod arrays of ZSM-5 directly grown on the ceramic honeycomb monolith were demonstrated with an advantage for faster transport of oligomerization reactants in and products out of the porous zeolites. An obvious mitigation of long-chain HCs and coking formation was accomplished in the multifunctional HC sorption/oxidation devices. This could be a generic merit for such array structured microporous film design as solutions to overcome these materials' constriction in sorption and reaction kinetics of small molecules in many anisotropic solid porous materials used in chemical and petrochemical production industries. The results and discussions of this work have been well presented with sound details. The work could be accepted after addressing the comments below.

1. The C-oriented array design is well versed, with sound proof and clearly-advantageous demo in the preferred fast transient sorption and reaction kinetics to benefit the adsorption, desorption, and conversion functions that may be involved in the multifunctional devices. How about other orientations? It would be helpful to discuss further with respect to the choices of guest molecules and host absorber structure design at the end.

Response: We thank the reviewer for the comment. Accordingly, we have included the following discussion in the revised manuscript.

“ZSM-5, as a versatile and abundant material, has uniform sub-nanometer-sized pore channels, tunable composition and acidic properties, and tailorable morphology. It is widely used in the chemical and petrochemical industries as selective catalysts, adsorbents, and membranes. Depending on the applications and target reactants, e.g., selective adsorption and separation for gases, catalytic reforming (methanol to gasoline, fluid catalytic cracking, and alkylation) or filtration (heavy metal and dye removal) etc., manipulating the ZSM-5 structures is vital to achieve a desirable performance. This is especially true considering the anisotropic pore distribution of ZSM-5. The anisotropic channel of ZSM-5 plays an essential role in determining the catalytic properties besides the acid properties and micropore size and geometry⁵⁴. ZSM-5 contains two types of interconnected 10-membered ring channels: the straight channel along the *b*-axis (5.6×5.3 Å) and the sinusoid one along the *a*-axis (5.5×5.1Å). NMR studies⁵⁵ have shown that diffusion in the direction of the *a*-axis of the crystals is much slower than in the *b*-axis direction. Typically, decreasing the *b*-axis channel length could shorten the path length of molecular diffusion and enhance accessibility to internal acid sites, which could enhance the activity of the catalyst in the acid-catalyzed reaction and reduce deactivation. Therefore, most of the literature reported work focuses on controlling ZSM-5 *b*-axis oriented growth to achieve a very thin thickness along *b*-axis, ranging from several nanometers to hundreds of nanometers^{54, 55, 56, 57}. Guest molecules such as methanol, benzene, *n*-heptane, and phenol etc. for alkylation and MTX catalytic reactions have been used to showcase the superior catalyst lifetime and activity. While limited work is available in terms of the *a*- and *c*-axes orientation control. Therefore, current work fills the gap and points out a possible route to push the ZSM-5 catalytic applications toward fast kinetics and high capacity. It is noted that the orientation control strategy is seldomly utilized for gas separation applications. Instead, tuning of the Al/Si ratio⁵⁸, pore modification⁵⁹ and ion exchange^{60, 61}, etc. approaches, based on a membrane configuration, are typically applied to achieve improved gas permeability and selectivity. Considering the smaller zig-zag channels along *a*-axis which can provide extended zeolite-guest molecules interaction, controlling of *a*-axis growth could offer a promising pathway to achieve improved performance for gas separation.”

2. Fig. 2 as of now is a bit unclear in the labelling, especially in (b) and (c), where the use of 57.6% and ratio labels is a bit confusing, similarly the units in the Table 1 are unclear due to the tabulate format issue. The authors should consider revise and improve the clarity of these data presentations throughout.

Response: We thank the reviewer for the comment. We have re-formatted the Figure 2 labelling and updated the Table 1 units.

3. Normally 'symbolic t' is used for the time parameter nomenclature wise, but 't' is directly used in the manuscript, the authors may consider adopt 'symbol t' instead.

Response: We thank the reviewer for the comment. We have re-formatted the manuscript carefully and thoroughly. The 'symbolic t' for the time parameter is adopted. We also adopted the 'symbolic T' for the temperature parameter.

4. There are quite some wording and grammar issues the authors should take care in order to make the flow better. For instance, the 'HZ-20-A' is used in singular tense mostly in the manuscript, but in some occasions plural tense is used. For example, in the discussion paragraphs over Fig. 2, it was spotted '... HZ-20-A yield less CO₂ and lower percentage of total generated CO₂ above 550 °C HZ-80-F sample'. In another occasion, 'The enrichment of captured C₃H₆..., which was also benefit to the adsorption process. However, ...if they could not be timely diffused out due to a potential diffusion barrier from zeolite micropores and block the accessibility to the acid sites.' The authors could make corrections here, say 'beneficial' in place of 'benefit', and '...could not timely diffuse out...' in place of '...could not be timely diffused out...'.

Response: We thank the reviewer for the comment. We have reviewed the manuscript thoroughly and addressed the wording and grammar issues.

5. Some typos and errors in the manuscript should be corrected, such as the reference error messages seemly occurred during the formatting.

Response: We thank the reviewer for the comment. We have reviewed the manuscript thoroughly to ensure all the links to the Figures/Tables/References are correct.

6. What is the next step given the novel design in the array structured absorber films? Some in-depth discussions over the structure arrangement, spacing, dimensions, and even the compositions, would be really worthwhile to help point to broadly engaged directions to the readers.

Response: We thank the reviewer for the comment. In this study, enhanced reaction kinetics and absorption capacity have been demonstrated for propene abatement with the novel array structured ZSM-5. The merits can be broadly applied and beneficial for extensive applications where reaction kinetics and absorption capacity are the bottlenecks. Those applications could be for bifunctional catalysts formulation for CO₂ capture and conversion, clean energy conversion and storage, VOCs enrichment and abatement, and downstream chemical and fuel processing, etc., where zeolite film morphology plays an important role. Or the applications could be for solid-solid interface reactions where increased contact points and active sites are essential, such as soot oxidation and waste plastic upcycling. Undoubtedly, the nanoarray structure parameters, such as space arrangement, array spacing, and orientation need to be carefully tuned based on the applications before an optimum performance can be achieved. In the meantime, the structure also provides many flexibilities by which the reaction performance could be easily manipulated. We have revised our manuscript accordingly to reflect the main points described here.

References:

1. Ma Q, Fu T, Ren K, Li H, Jia L, Li Z. Controllable Orientation Growth of ZSM-5 for Methanol to Hydrocarbon Conversion: Cooperative Effects of Seed Induction and Medium pH Control. *Inorganic Chemistry* **61**, 13802-13816 (2022).
2. Zhang J, *et al.* Tailored Synthesis of ZSM-5 Nanosheets with Controllable b-Axis Thickness and Aspect Ratio: Strategy and Growth Mechanism. *Chemistry of Materials* **34**, 3217-3226 (2022).
3. Sun Y, *et al.* Fabrication of Twin-Free Nanoslab ZSM-5 Zeolite with b-Axis Orientation for Super MTP Catalyst. *ACS Sustainable Chemistry & Engineering* **10**, 9431-9442 (2022).
4. Zhang J, *et al.* b-Axis-Oriented ZSM-5 Nanosheets for Efficient Alkylation of Benzene with Methanol: Synergy of Acid Sites and Diffusion. *ACS Catalysis* **13**, 3794-3805 (2023).
5. Kwan SM, Leung AYL, Yeung KL. Gas permeation and separation in ZSM-5 micromembranes. *Separation and Purification Technology* **73**, 44-50 (2010).
6. Flanders CL, Tuan VA, Noble RD, Falconer JL. Separation of C6 isomers by vapor permeation and pervaporation through ZSM-5 membranes. *Journal of Membrane Science* **176**, 43-53 (2000).
7. Aoki K, Tuan VA, Falconer JL, Noble RD. Gas permeation properties of ion-exchanged ZSM-5 zeolite membranes. *Microporous and Mesoporous Materials* **39**, 485-492 (2000).
8. Xiong R, *et al.* Hydrogen isotopes separation in Ag(I) exchanged ZSM-5 zeolite through strong chemical affinity quantum sieving. *Microporous and Mesoporous Materials* **313**, 110820 (2021).

REVIEWERS' COMMENTS

Reviewer #1 (Remarks to the Author):

I believe the authors have effectively addressed all the concerns I raised during my previous review. In particular, the additional synthesis work related to Si/Al ratios and Na⁺ is thorough and offers valuable insights into the crystallization process. As one of the reviewers noted, the use of IR may not be entirely convincing due to the limitations of the DRIFT technique in characterizing the samples with monolith structure, a common constraint in studies with specific applications. I hope that in the future, the authors can enhance the understanding further by employing characterization techniques better suited for such monoliths. Overall, I recommend publishing the manuscript.

Reviewer #2 (Remarks to the Author):

Most of my comments and questions were addressed in detail by the authors and the effort is appreciated.

Nevertheless, I'm still not totally convinced by the interpretation of DRIFTS data and breakthrough curves. On the one hand, there are some bands assignments by DRIFTS that still lack accuracy or are even misleading (lines 249-251, assignment to stretching vibrations ?????). On the other hand, and according to the reviewer's own experience, the very strong absorption of zeolite framework itself is a problem for obtaining reliable DRIFT data. Some parts of the IR spectrum could be not visible and bands could be distorted because their intensities is also function of the specific absorption of the material in the different spectral regions.

It is recommended to not over-emphasize all along the whole manuscript the role played by film structuration because the two investigated materials have also different chemical properties (amount and type of acid sites and different Si/Al ratio). This makes the conclusions less obvious than if the materials would be similar in composition.

Otherwise, there are still remaining some broken reference links all along the manuscript which need to be corrected.

Reviewer #4 (Remarks to the Author):

The authors have addressed my comments, the revised manuscript could be considered for publication now.

Point-to-point Responses to Reviewers' Comments

Responses to Reviewer #1:

I believe the authors have effectively addressed all the concerns I raised during my previous review. In particular, the additional synthesis work related to Si/Al ratios and Na⁺ is thorough and offers valuable insights into the crystallization process. As one of the reviewers noted, the use of IR may not be entirely convincing due to the limitations of the DRIFT technique in characterizing the samples with monolith structure, a common constraint in studies with specific applications. I hope that in the future, the authors can enhance the understanding further by employing characterization techniques better suited for such monoliths. Overall, I recommend publishing the manuscript.

Response: We thank the reviewer for the strong endorsement of this work and our efforts to further improve the manuscript. Ideally it would be better to exclude the influence of monolithic substrates for DRIFT characterization. However, grinding the samples into powders will destroy the secondary-grown array structure, which is the unique characteristic of our proposed structure design. We will continue to work on more comprehensive characterizations and new characterization technique development for the monolithic adsorbers, catalysts, and other devices in the future work. In recognition of these needs of future study built on this work, we add a paragraph below in the conclusion section of the further revised manuscript.

'It is worth noting that to reveal exact structure-function relationships, some important questions remain open on the heterogeneous adsorbers and catalysts with ZSM-5 nanorod arrays demonstrated here. For example, further *in-situ/Operando* structure characterizations and characterization technique development are needed for decoupling the sorption and reaction functions of individual constituents in such a multi-component structured device during operation. These new characterization efforts will help unravel the role monolithic substrate plays on the sorption behavior of ZSM-5 nanorod array, complementing to the *in situ* DRIFTS study in this work. In addition, refining synthesis control of array-structured ZSM-5 film on monoliths and understanding its exact growth mechanism can help identify and understand the structure (e.g., crystallite size, intercrystallite spacing) and chemical composition (e.g., Si/Al ratio) effects on the sorption and reaction kinetic performance.'

Responses to Reviewer #2:

Most of my comments and questions were addressed in detail by the authors and the effort is appreciated. Nevertheless, I'm still not totally convinced by the interpretation of

DRIFTS data and breakthrough curves. On the one hand, there are some bands assignments by DRIFTS that still lack accuracy or are even misleading (lines 249-251, assignment to stretching vibrations ???). On the other hand, and according to the reviewer's own experience, the very strong absorption of zeolite framework itself is a problem for obtaining reliable DRIFT data. Some parts of the IR spectrum could be not visible and bands could be distorted because their intensities is also function of the specific absorption of the material in the different spectral regions.

Response: We thank the reviewer for the insightful comments. It is very challenging to assign the observed DRIFTS bands as both ZSM-5 and cordierite monolith were involved during the characterization process. On one hand, we observed non-conventional DRIFTS bands for C₃H₆ at the wavenumber range lower than 1400 cm⁻¹, which might be impacted by the influence of ZSM-5 strong absorption. As the observed peaks repeatedly present over the adsorption and desorption process, we did believe the observed peaks were true and not caused by measurement error. On the other hand, the methodology to measure the change of adsorbed H-C₃H₆ (~1470 cm⁻¹) and oligomeric species (3080-2750 cm⁻¹) via integrating the intensity of corresponding DRIFTS peaks were adopted from the literature¹. Therefore, to avoid any confusion or misleading, we have taken down the spectra at the wavenumber range lower than 1400 cm⁻¹ and focus on the analysis on the peak at ~1470 cm⁻¹ using the adopted method. The manuscript and DRIFTS spectra in Supplementary Figure 9 and Supplementary Figure 10 are updated accordingly.

It is recommended to not over-emphasize all along the whole manuscript the role played by film structuration because the two investigated materials have also different chemical properties (amount and type of acid sites and different Si/Al ratio). This makes the conclusions less obvious than if the materials would be similar in composition.

Response: We thank the reviewer for the suggestions. Over the manuscript we have highlighted that the unique ZMS-5 array structure is clearly distinguishable from the other conventional dense film, resulting in superior advantages in the reaction dynamics. The results of DRIFTS area integration methodology could be considered as indirect support for our hypothesis. We do admit that the chemical properties resulted from the different Si/Al ratios between two samples might play roles for performance improvement. We will continue to work on the recipe refinement to synthesize ZSM-5 samples where the impact of chemical properties could be excluded. We have revised the manuscript accordingly. Also, in recognition of the characterization and synthesis efforts needed in the future study, we add a paragraph below in the conclusion section of the further revised manuscript.

'It is worthing noting that to reveal exact structure-function relationships, some important questions remain open on the heterogeneous adsorbers and catalysts with ZSM-5

nanorod arrays demonstrated here. For example, further in-situ/Operando structure characterizations and characterization technique development are needed for decoupling the sorption and reaction functions of individual constituents in such a multi-component structured device during operation. These new characterization efforts will help unravel the role monolithic substrate plays on the sorption behavior of ZSM-5 nanorod array complementing to the *in situ* DRIFTS study in this work. In addition, refining synthesis control of array-structured ZSM-5 film on monoliths and understanding its exact growth mechanism can help identify and understand the structure (e.g., crystallite size, intercrystallite spacing) and chemical composition (e.g., Si/Al ratio) effects on the sorption and reaction kinetic performance.'

Otherwise, there are still remaining some broken reference links all along the manuscript which need to be corrected.

Response: We thank the reviewer for the comment. We realize that the broken reference links refer to the supplementary figures and tables in the Supplementary file. Accordingly, we have addressed all the broken links in the latest revision.

Reviewer #4 (Remarks to the Author):

The authors have addressed my comments, the revised manuscript could be considered for publication now.

Response: We thank the reviewer for the strong endorsement of this work and our efforts to address the reviewers' comments.

Reference

1. Bernauer M, *et al.* Proton proximity – New key parameter controlling adsorption, desorption and activity in propene oligomerization over H-ZSM-5 zeolites. *Journal of Catalysis* **344**, 157-172 (2016).